# Root Cause Analysis of Failures in Microservices
# via Bayesian Root Cause Discovery

**Kenneth Lee** [1]   **Zihan Zhou** [2]   **Murat Kocaoglu** [2]

## Abstract

Modern cloud systems rely on architectures with many interconnected microservices, which enable scalability and flexibility but make troubleshooting failures difficult. Identifying the root cause requires navigating complex dependencies, often beyond the capacity of domain experts. Causal models offer a principled approach to root cause analysis (RCA), but prior methods are typically sample inefficient, as they assume access to the full causal graph or require large numbers of post-failure interventions. We introduce Bayesian Root Cause Discovery (BRCD), which leverages a partial causal structure (a CPDAG learned during the pre-failure period) and performs Bayesian inference without enumerating all DAGs from each interventional Markov equivalence class ($\mathcal{I}$-MEC) for each root cause candidate. Using a recent uniform DAG sampling framework (Wienöbst et al., 2023), BRCD provides the first statistical consistency guarantees for nonparametric RCA, with both identifiability and finite-sample posterior bounds under $\varepsilon$-vanishing approximation. Empirically, across synthetic benchmarks and three microservice systems (Online Boutique, Sockshop, Petshop), BRCD achieves state-of-the-art top-$l$ accuracy while remaining effective in low-failure-sample regimes and scaling to large graphs. Our code is available at https://github.com/kenneth-lee-ch/brcd.

[1]Elmore Family School of Electrical and Computer Engineering, Purdue University, West Lafayette, IN, USA [2]Department of Computer Science, Johns Hopkins University, Baltimore, MD, USA. Correspondence to: Kenneth Lee <lee4094@purdue.edu>, Zihan Zhou <zzhou150@jh.edu>, Murat Kocaoglu <mkocaoglu@jhu.edu>.

*Proceedings of the 43rd International Conference on Machine Learning*, Seoul, South Korea. PMLR 306, 2026. Copyright 2026 by the author(s).

## 1. Introduction

Root cause analysis (RCA) seeks to pinpoint the underlying source of a system's abnormal behavior. It plays a vital role in system reliability across various domains, including manufacturing (Schmidt et al., 2019; Oliveira et al., 2023), cloud applications (Zhang et al., 2021; Żurkowski & Zieliński, 2024), IT operations (Soldani & Brogi, 2022), telecommunication networks (Zhang et al., 2020; Chen et al., 2022), and healthcare (Uberoi et al., 2007). For example, manufacturing companies can experience a hundred different production disturbances due to equipment failure, human error, waiting time for materials, and subsequent stops in output flow from machines on a daily basis (Soares Ito et al., 2022). Effective diagnosis of the immediate causes behind these disturbances is an integral part of a resilient production system to prevent financial losses and increase efficiency (Ito et al., 2022). A similar issue also occurs in cloud applications. Cloud applications typically follow a microservices-based architecture to perform different subtasks for the benefits of scalability and greater flexibility (Soldani et al., 2018; El Akhdar et al., 2024). Although microservices offer numerous advantages, an anomaly in one service can trigger a chain reaction due to the interdependencies between services, potentially causing additional issues and leading to system failure (Liu et al., 2016).

With the continuous growth of system operations, microservices can form complex interconnected networks (Gao et al., 2014; Amini et al., 2020). Particularly, these microservices typically come with Key Performance Indicators (KPIs) such as latency and metric data like CPU or memory utilization for monitoring their states and availabilities in real-time. The complexity of the system and the large number of metrics make root cause analysis a daunting task for engineers, hindering their ability to identify the cause of failures promptly. This motivates the use of causal discovery for RCA in microservices (Chen et al., 2014; Lin et al., 2018; Qiu et al., 2020; Gan et al., 2021; Li et al., 2022; Ikram et al., 2022; Wang et al., 2023; Xin et al., 2023). A typical causal discovery-based RCA approach begins by constructing a graph, often represented as a directed acyclic graph, also known as a causal graph. In this graph, each node represents a metric, and a directed edge between two

nodes indicates a causal relationship. System engineers designate a KPI, also known as the trigger point, to alert the engineers when the system becomes anomalous (Lin et al., 2024). The final step typically involves applying scoring methods to rank potential root causes among all the metrics in the graph, identifying those most likely responsible for the compromised system performance.

While many RCA in microservices focus on accurately locating the root cause, the sample efficiency is also crucial. For instance, an hour-long outage of Amazon Web Services can result in a loss of millions of dollars in revenue (Kaiser, 2017). Therefore, every sample of a malfunction in a system can be expensive to collect and should be limited. Many existing RCA methods that rely on a formal causal methodology use a constraint-based approach. Constraint-based methods form the foundations of classical causal discovery algorithms such as PC algorithm (Spirtes et al., 2001). Given the observational distribution, one can learn the causal graph up to its Markov equivalence class (MEC). These are extended to the interventional setting, which is adapted into the root-cause discovery (RCD) algorithm by Ikram et al. (2022) for solving the RCA problem. Therefore, they inherit the challenges faced by constraint-based algorithms: Mistakes due to finite samples can propagate and affect the algorithm significantly later. Even though the RCD algorithm provides some relief to this issue by hierarchically solving smaller problems without learning the causal graph completely, it sequentially applies sample-inefficient invariance tests, in which statistical power decreases drastically as the conditioning set size increases.

Bayesian approaches, on the other hand, do not suffer from these issues that affect constraint-based methods. They optimally incorporate the available data to find the MAP estimate. For causal discovery, Bayesian approaches have been challenging due to super-exponential size of the number of causal graphs. The classical literature had to make parametric assumptions to sidestep iterating over this large space. Recently, Zhou et al. (2024) proposed an efficient Bayesian causal discovery algorithm relying on the breakthrough result by Wienöbst et al. (2023) that shows we can sample from a Markov equivalence class of causal graphs in polynomial time with respect to the number of variables .

We start by observing that an ample amount of observational monitoring data is usually available for a system before a failure occurs. These data can be used to obtain a partial causal structure via causal discovery from observational data (Ikram et al., 2025). Recently, Ikram et al. (2022) modeled the system failure as an intervention, which can be represented with a binary variable that points to the root cause on the ground truth causal graph. We observe that we can leverage a connection between the RCA problem and the notion of interventional Markov equivalence class ($\mathcal{I}$-

MEC) to help reduce the search space for Bayesian inference (Yang et al., 2018; Kocaoglu et al., 2019; Jaber et al., 2020; Zhou et al., 2025). Our key observation is that *all causal graphs in the same $\mathcal{I}$-MEC entail the same data likelihood*. Hence, we can group super-exponentially many DAGs into a single DAG likelihood evaluation per $\mathcal{I}$-MEC. This avoids enumerating all DAGs in each $\mathcal{I}$-MEC to achieve efficient posterior updates . Furthermore, partitioning different $\mathcal{I}$-MEC can be done in a straightforward manner by orienting the neighborhood of each root cause candidate.

Upon these observations, we propose a novel Bayesian root cause analysis algorithm, namely **Bayesian Root Cause Discovery (BRCD)**. **BRCD** can localize the root causes without knowing the ground truth causal structure and provide a principled uncertainty quantification to measure which metric is anomalous. It is sample-efficient as it does not rely on independence tests, making it particularly well-suited for identifying root causes with limited data. Second, it avoids relying on parametric assumptions, allowing for greater flexibility and generalizability across diverse systems. Also, the Bayesian method has an *anytime* property. It enables iterative updates to the posterior distributions of potential root causes as more data becomes available, facilitating the identification of the most likely candidates at any stage. We summarize our contributions as follows:

- We propose the first Bayesian root cause discovery method that leverages a polynomial-time DAG sampling method to directly identify the root cause via posterior approximation given a partial causal structure in the form of a completed partially directed acyclic graph (CPDAG) .

- We provide the first theoretical guarantees for nonparametric RCA: (i) the identifiability of root causes under extended faithfulness when only a CPDAG is available (Lemma 4.1), and (ii) posterior consistency with an exponential finite-sample bound that remains valid even when the likelihood is computed using an $\epsilon$-accurate plug-in estimator (Theorem 4.3, 4.4).

- **BRCD** outperforms the existing state-of-the-art methods in our synthetic experiments and in real-world applications such as Online Boutique, Sockshop, and Petshop (Hardt et al., 2023). **BRCD** remains accurate even with very few anomalous samples, leveraging pre-failure observational data.

**Paper Outline:** Section 2 reviews the background on causal Bayesian networks and the role of interventions. Section 3 formalizes the RCA setting for microservices and how we model it with causal graphs. Section 4 presents **BRCD** along with theoretical analysis. Section 5 details the experimental setup and reports results on both synthetic

and real-world datasets. Section 6 discusses limitations and future directions. Related work is deferred to the appendix, which also contains detailed proofs, implementation notes, additional experiments, and extended discussions.

## 2. Background

**Causal Bayesian Networks**   A Bayesian network represents a set of random variables $\mathbf{X} = \{X_i\}_{i \in [d]}$ and their conditional dependencies using a directed acyclic graph (DAG) $\mathcal{G}$ . The joint distribution $p(\mathbf{X})$ can be factorized as

$$p(\mathbf{X}) = \prod_i p(X_i | Pa_{\mathcal{G}}(X_i)) \tag{1}$$

where $Pa_{\mathcal{G}}(X_i)$ is the set of parents of $X_i$ in $\mathcal{G}$. We omit the subscripts when the graph is clear from the context. Causal Bayesian networks are used to define a causal model that specifies the observational and interventional distributions via the truncated factorization formula (Pearl, 2009). We denote the ground truth DAG as $\mathcal{G}^{\star}$.

**d-separation**   In a DAG $\mathcal{G} = (\mathbf{V}, \mathbf{E})$, $\langle X, Y, Z \rangle$ is called a collider if $X \to Y \leftarrow Z$ is in $\mathcal{G}$. Otherwise, it is called a non-collider in $\mathcal{G}$. A path $\pi$ between vertices $X$ and $Y$ is *d-connecting (active)* relative to a set of vertices $\mathbf{Z}$ ($X, Y \notin \mathbf{Z}$) if $(i)$ every non-collider on $\pi$ is not in $\mathbf{Z}$ and $(ii)$ every collider on $\pi$ is an ancestor of some $Z \in \mathbf{Z}$. If there is no d-connecting path between $X$ and $Y$ relative to $\mathbf{Z}$, we say $X$ and $Y$ are *d-separated* relative to $\mathbf{Z}$.

**Global Markov property**   Let $p$ be a distribution that factorizes according to a Bayesian network with the structure $\mathcal{G} = (\mathbf{V}, \mathbf{E})$. Let $\mathbf{X}, \mathbf{Y}, \mathbf{Z} \subset \mathbf{V}$ be three disjoint sets of variables. If $\mathbf{X}$ and $\mathbf{Y}$ are d-separated by $\mathbf{Z}$, then $\mathbf{X}$ is conditionally independent of $\mathbf{Y}$ given $\mathbf{Z}$.

**Markov Equivalence and CPDAGs**   Two DAGs $\mathcal{G}_1, \mathcal{G}_2$ with the same set of vertices are *Markov equivalent* if for any three disjoint set of vertices $\mathbf{X}, \mathbf{Y}, \mathbf{Z}$, $\mathbf{X}$ and $\mathbf{Y}$ are d-separated by $\mathbf{Z}$ in $\mathcal{G}_1$ if and only if $\mathbf{X}$ and $\mathbf{Y}$ are d-separated by $\mathbf{Z}$ in $\mathcal{G}_2$. The set of all DAGs that encode the same set of conditional independence as the DAG $\mathcal{G}$ induced only by the global Markov property is called the *Markov equivalence class* of $\mathcal{G}$ , denoted as $[\mathcal{G}]$. A Markov equivalence class (MEC) can be compactly represented by a *completed partially directed acyclic graph* (CPDAG). A directed edge $X \to Y$ is in a CPDAG if and only if $X \to Y$ is in $\mathcal{G}' \in [\mathcal{G}]$ for all $\mathcal{G}'$ . An undirected edge $X - Y$ in a CPDAG implies there is a DAG $\mathcal{G}_1$ that has $X \to Y$ and another DAG $\mathcal{G}_2$ has $X \leftarrow Y$, where $\mathcal{G}_1, \mathcal{G}_2 \in [\mathcal{G}]$. Generally, one can learn up to a MEC of DAGs from an observational distribution under faithfulness assumption (Andersson et al., 1997; Spirtes et al., 2001). A CPDAG is also referred to as the essential graph $\mathcal{E}(\mathcal{G})$ of the original graph.

**Interventions**   An intervention on a variable $X$ corresponds to replacing the original conditional distribution $p(X|Pa(X))$ with a different distribution $p'(X|Pa(X))$. An intervention is considered *soft* if the variable remains dependent on its parents and *hard* otherwise. For a DAG $\mathcal{G} = (\mathbf{V}, \mathbf{E})$, given a set of intervened variables $\mathbf{X} \subseteq \mathbf{V}$, the interventional distribution factorizes as follows

$$p_{\mathbf{X}}(\mathbf{V}) = \prod_{\{i | X_i \in \mathbf{X}\}} p'(X_i | Pa(X_i)) \prod_{\{j | X_j \notin \mathbf{X}\}} p(X_j | Pa(X_j)) \tag{2}$$

Note that the conditional distributions of the variables that are not intervened on do not change with respect to the observational distribution. This is often called distributional invariance (Peters et al., 2016). Graphically, a hard intervention on $X$ severs the incoming edges to $X$ whereas a soft intervention leaves these edges intact. A proxy variable $F$ has been used to represent an intervention by augmenting the ground truth DAG $\mathcal{G}^{\star}$ (Pearl, 1995; Yang et al., 2018).

**$\mathcal{I}$-Markov Equivalence Class**   If, in addition to the observational distribution, we have access to a set of interventional distributions under hard interventions on sets of variables  $\mathcal{I}$, the edges induced by the intervention targets can be oriented using independence tests. With further refining by applying Meek Rules (Meek, 1995), we can recover a *maximally partially directed acyclic graph* (MPDAG). As demonstrated by Hauser & Bühlmann (2012), MPDAGs are also chain graphs with chordal components. An MPDAG characterizes the $\mathcal{I}$-Markov equivalence class ($\mathcal{I}$-MEC) and is referred to as the $\mathcal{I}$-essential graph or $\mathcal{I}$-CPDAG (since MPDAGs are valid CPDAGs), denoted as $\mathcal{E}_{\mathcal{I}}(\mathcal{G})$ or $\mathcal{C}_{\mathcal{I}}(\mathcal{G})$. Two causal graphs are considered $\mathcal{I}$-Markov equivalent with respect to an intervention set $\mathcal{I}$ if they share the same $\mathcal{I}$-essential graph. We now introduce the concept of cut configurations. A crucial connection between cut configurations and $\mathcal{I}$-MECs is that cut configurations helps us to distinguish various $\mathcal{I}$-CPDAGs . For each target set $I \in \mathcal{I}$, we define the collection of all possible orientations of edges in $\mathbf{E}[I, \mathbf{V} \setminus I]$ as $Q(I)$, where $\mathbf{E}[I, \mathbf{V} \setminus I]$, called edge cut on $I$, is set of all edges with one end in $I$ and the other end in $\mathbf{V} \setminus I$. The possible orientations of edge cuts on $I$ are called cut configurations. We denote the $k$-th cut configuration as $Q_k[I], k \in [n_I]$ where $n_I$ is the total number of possible cut configurations on $I$. We denote the $\mathcal{I}$-CPDAG that has $Q_k(I)$ on $I$ as $\mathcal{C}_{I_k}(\mathcal{G})$.

**Assumption 2.1** (Interventional faithfulness). Let $\mathcal{G} = (\mathbf{V}, \mathbf{E})$ be a DAG and let $R \subseteq \mathbf{V}$ be a set of root cause variables. Let $\mathcal{G}_{aug} = (\mathbf{V} \cup \{F\}, \mathbf{E} \cup \{F \to v : v \in R\})$. Let $p$ be the joint distribution over $\mathcal{G}_{aug}$. We say that $p$ is *interventionally faithful* to $\mathcal{G}_{aug}$ if for all disjoint $\mathbf{X}$ and $\mathbf{Y}$ in $\mathbf{V}$, $\mathbf{X}$ and $\mathbf{Y}$ are d-separated by $\mathbf{Z}$ in $\mathcal{G}_{aug}$ if and only if $\mathbf{X}$ and $\mathbf{Y}$ are conditional independent given $\mathbf{Z}$ in $p$.

# 3. Problem Formulation

Consider a microservice system that consists of $N$ services $\{w_i\}_{i=1}^N$. Before a failure occurs, the monitoring system collects $W$ many metrics from each service $i$ up to a time step $T$ to form an observational dataset $\mathcal{D}_{obs} = \{\mathbf{X}_t^i \mid t \in \{1, \ldots, T\}, i \in \{1, \ldots, N\}\}$, where $\mathbf{X}_t^i = \{X_t^{(i,j)}\}_{j=1}^W$. Suppose a failure occurs at time step $T+1$ and the monitoring system continues to collect the metrics from each service up to some time step $\lambda$ to form another dataset $\mathcal{D}_{int} = \{\mathbf{X}_t^i \mid t \in \{T+1, \ldots, \lambda\}, i \in \{1, \ldots, N\}\}$.

Recently, Ikram et al. (2022) models a failure in a microservice system as a soft intervention. Particularly, since the time of failure occurrence is usually available, Ikram et al. (2022) utilizes a proxy variable $F$ as a binary indicator variable to differentiate the monitoring data collected before and after a failure for RCA. By combining $\mathcal{D}_{obs}$ and $\mathcal{D}_{int}$, one can sample from the distribution $p^\star$, where $p^\star(.|F=0)$ is the observational distribution and $p^\star(.|F=1)$ is the interventional distribution. Particularly, this observation allows us to model the data-generating mechanism of the combined dataset $\mathcal{D}$ as a DAG augmented by the proxy variable $F$, which we will call $\mathcal{G}_{aug}^\star$ throughout this work, where $F$ is the parent of the root cause in $\mathcal{G}_{aug}^\star$. We provide Figure 1 to illustrate the idea.

In this work, we follow Ikram et al. (2022) to assume the data to be identically and independently distributed. We consider the setting in which one has learned a CPDAG $\mathcal{C}(\mathcal{G}^\star)$ from $\mathcal{D}_{obs}$ via observational causal discovery prior to the occurrence of failure. A more detailed discussion on observational causal discovery algorithms can be found in Appendix A. Our objective is to identify the metric variable that is the root cause, which is the child of $F$ in the unknown $\mathcal{G}_{aug}^\star$. We assume no latent variables. We also consider the setting where multiple root causes are present.

# 4. Methodology

In this section, we describe our main algorithm and its theoretical guarantees in identifying the root causes. The overall framework is shown in Figure 1. The proposed algorithm is presented in Algorithm 2. All the proofs are provided in Appendix B. The core contribution of this work is to draw on an efficient graph sampling scheme for posterior update to devise a sample-efficient RCA algorithm. We will first discuss the setting where a ground truth CPDAG is given as an input. Then, we will discuss a version of our algorithm to handle the uncertainty of the CPDAGs.

**Bayesian Inference of the Root Causes** Given the combined normal and anomalous monitoring dataset $\mathcal{D} = \mathcal{D}_{obs} \cup \mathcal{D}_{int}$ and a CPDAG $\mathcal{C}(\mathcal{G}^\star)$ for some ground truth DAG $\mathcal{G}^\star = (\mathbf{V}, \mathbf{E})$, Bayesian inference over root causes aims to estimate the full posterior probability density over

---

**Algorithm 1 Augmented Graphs Sampling**

**input** a CPDAG $\mathcal{C}(\mathcal{G}^\star)$, where $\mathcal{G}^\star = (\mathbf{V}, \mathbf{E})$, number of root causes $k$
**output** A set of augmented DAGs $\mathcal{S}_{\mathcal{G}_{aug}}$
1: $\mathcal{S}_{\mathcal{G}_{aug}} = \emptyset, T = 0$
2: **for** $\mathbf{R} \subseteq \mathbf{V}, |\mathbf{R}| = k$ **do**
3:     Add $F \to \mathbf{R}$ to create $\mathcal{G}_p$ from $\mathcal{C}(\mathcal{G}^\star)$
4:     **for** each cut configuration of the neighborhood of $\mathbf{R}$ **do**
5:         $\mathcal{C}_{\mathbf{R}}(\mathcal{G}^\star) \leftarrow$ Apply Meek rules (Meek, 1995) to $\mathcal{G}_p$
6:         $\mathcal{G}_{aug}, Q_i \leftarrow$ Sample a DAG from $\mathcal{C}_{\mathbf{R}}(\mathcal{G}^\star)$ and compute the size of $\mathcal{C}_{\mathbf{R}}(\mathcal{G}^\star)$ via the clique-picking algorithm (Wienöbst et al., 2023)
7:         Add $(\mathcal{G}_{aug}, Q_i)$ to $\mathcal{S}_{\mathcal{G}_{aug}}$
8: **return** $\mathcal{S}_{\mathcal{G}_{aug}}$ and $T = \sum_i Q_i$

---

**Algorithm 2 BRCD**

**input** Observational and interventional data $\mathcal{D}$, a prior of the root causes $p(\mathbf{R})$, a CPDAG $\mathcal{C}(\mathcal{G}^\star)$.
**output** A posterior distribution $p(\mathbf{R}|\mathcal{D})$
1: $\mathcal{S}_{\mathcal{G}_{aug}}, T \leftarrow$ Apply Algorithm 1 to $\mathcal{C}(\mathcal{G}^\star)$.
2: **for** $\mathbf{R}' \subseteq \mathbf{V}$ **do**
3:     **for** $(\mathcal{G}_{aug}, Q_i) \in \mathcal{S}_{\mathcal{G}_{aug}}$, where $F \to \mathbf{R}'$ in $\mathcal{G}_{aug}$ **do**
4:         Compute $p(\mathcal{D}|\mathcal{G}, \mathbf{R}') \leftarrow p(\mathbf{V}, F)$ according to $\mathcal{G}_{aug}$ by the product $\prod_{X \in \mathbf{V} \cup F} p(X|Pa(X))$
5:         $M_i = p(\mathcal{D}|\mathcal{G}, \mathbf{R}')p(\mathcal{G}|\mathbf{R}')$ where $p(\mathcal{G}|\mathbf{R}') = Q_i/T$
6:     $p(\mathcal{D}|\mathbf{R}') \leftarrow \sum_i M_i$
7: $p(\mathbf{R}|\mathcal{D}) \leftarrow p(\mathcal{D}|\mathbf{R})p(\mathbf{R})/(\sum_{\mathbf{R}'} p(\mathcal{D}|\mathbf{R}')p(\mathbf{R}'))$
8: **return** $p(\mathbf{R}|\mathcal{D})$

---

BNs that model the observations. Given that any metric can be the root cause, we simply have a uniform prior over all the metrics e.g. $p(\mathbf{R}) = 1/\binom{N}{k}$, where $k$ represents the number of metrics being the true root causes [1], $N$ is the number of observed variables, and $\mathbf{R}$ denotes the set of true root causes. Note that any prior belief on the potential metric being anomalous can easily be incorporated. Formally, the posterior of root causes can be approximated as follows

$$p(\mathbf{R}|\mathcal{D}) = \frac{p(\mathcal{D}|\mathbf{R})p(\mathbf{R})}{\sum_{\mathbf{R}'} p(\mathcal{D}|\mathbf{R}')p(\mathbf{R}')} \tag{3}$$

$$p(\mathcal{D}|\mathbf{R}) = \sum_{\mathcal{G} \in [\mathcal{G}^\star]} p(\mathcal{D}|\mathcal{G}, \mathbf{R})p(\mathcal{G}|\mathbf{R}) \tag{4}$$

and $p(\mathcal{G}|\mathbf{R}) = 1/T$ where $T$ is the total number of graphs in all $\mathcal{I}$-MECs represented by all possible $\mathcal{I}$-CPDAGs. At first, computing $p(\mathcal{D}|\mathbf{R})$ seems intractable. However, Wienöbst et al. (2023) shows that sampling a DAG from its MEC and counting the size of the MEC only takes polynomial-time with respect to the size of the graph. Hence, we can compute $p(\mathcal{D}|\mathbf{R})$ by sampling DAGs based on the given CPDAG $\mathcal{C}(\mathcal{G}^\star)$ learned during the normal operation time. Additionally, we draw a critical connection between modeling failure as a soft intervention and computing $p(\mathcal{D}|\mathcal{G}, \mathbf{R})-$ the true

---

[1]We accommodate the setup where one is interested in ranking $k$ variables together in RCA for multiple root causes.

interventional distribution factorizes according to $\mathcal{G}^\star_{aug}$ as in (2), where $\mathcal{G}^\star_{aug}$ is $\mathcal{G}^\star$ augmented with the edge $F \to \mathbf{R}$, and $\mathbf{R}$ is the set of true root causes. Hence, we can sample $\mathcal{G}_{aug}$ directly after sampling $\mathcal{G}$ based on $\mathcal{C}(\mathcal{G}^\star)$ by augmenting $\mathcal{G}$ with $F \to \mathbf{X}$ for some variables $\mathbf{X}$ in $\mathcal{G}$ according to the index values $\mathbf{R}$ takes to approximate $p(\mathcal{D}|\mathcal{G}, \mathbf{R})$. We provide an example via Figure 1 for a single root cause case.

Next, we discuss the identifiability results for our proposed method for finding multiple root causes.

**Lemma 4.1** (Generic Identifiability of multiple root causes). *Let $\mathcal{G} = (\mathbf{V}, \mathbf{E})$ be a DAG and let $\mathbf{R} \subseteq \mathbf{V}$ be a set of root causes. Under* modularity, *positivity (all relevant conditionals are strictly positive on their supports), and that the nodewise mechanisms are* non-degenerate *(for $V \in \mathbf{R}$ there exist parent values and two child values with different conditional likelihood ratios across $F$), then for almost all parameter values (excluding a measure-zero subset), any two distinct target sets $\mathbf{R} \neq \mathbf{R}'$ induce distinct interventional families $\{p(\mathbf{X} \mid F = f, \mathcal{G})\}_{f \in \{0,1\}}$. In particular, if $(\mathcal{G}^\star, \mathbf{R}^\star)$ is the ground truth, then*

$$\Delta_{\min} := \inf_{(\mathcal{G},\mathbf{R}) \neq (\mathcal{G}^\star,\mathbf{R}^\star)} \mathbb{E}_{p^\star}\left[\log \frac{p(\mathbf{X}|\mathcal{G}^\star,\mathbf{R}^\star)}{p(\mathbf{X}|\mathcal{G},\mathbf{R})}\right] > 0. \quad (5)$$

**Intuition of Lemma 4.1** Lemma 4.1 implies that, under interventional faithfulness, any incorrect augmentation $(\mathcal{G}, \mathbf{R}) \neq (\mathcal{G}^\star, \mathbf{R}^\star)$ incurs a strictly positive asymptotic log-likelihood gap (equivalently, a positive KL separation), so the marginal likelihood $p(\mathcal{D} \mid \mathbf{R})$ concentrates on the true intervention target set $R^\star$. To compute $p(\mathcal{D} \mid \mathbf{R})$ efficiently, we avoid scoring individual DAGs and instead score interventional Markov equivalence classes: all DAGs within the same $\mathcal{I}$-MEC represented by an $\mathcal{I}$-CPDAG induce the same interventional likelihood. Consequently, for each candidate $\mathbf{R}$ we enumerate all compatible $\mathcal{I}$-CPDAGs by orienting all possible cut configurations between $\mathbf{R}$ and $\mathbf{V} \setminus \mathbf{R}$ and closing under Meek rules, and then sample one DAG per $\mathcal{I}$-CPDAG to approximate $p(\mathcal{D} \mid \mathbf{R})$. This enumeration is exhaustive. Corollary 4.2 formalizes that Algorithm 1 generates all and only $\mathcal{I}$-CPDAGs compatible with $(\mathcal{C}(\mathcal{G}^\star), \mathbf{R})$.

**Corollary 4.2** (Completeness of Algorithm 1). *For any candidate target set $\mathbf{R}$, Algorithm 1 enumerates all and only the $\mathcal{I}$-CPDAGs compatible with $(\mathcal{C}(\mathcal{G}^\star), \mathbf{R})$ by ranging over all possible orientations of the edge cut $E[\mathbf{R}, \mathbf{V} \setminus \mathbf{R}]$ and closing under Meek rules. Hence Algorithm 1 covers all $\mathcal{I}$-MECs consistent with $\mathcal{C}(\mathcal{G}^\star)$ and $\mathbf{R}$.*

**Robustness to finite-sample approximation** We often evaluate likelihoods using a plug-in distribution $\hat{p}$ rather than the unknown truth $p^\star$. Here, we give the posterior consistency and finite-sample bounds of **BRCD** even when $\hat{p}$ is away from $p$. First, we make two assumptions besides faithfulness and causal sufficiency:

**(A1)** ($\varepsilon$-**close plug-in**). With probability at least $1 - \eta, \eta \in (0, 1), n = |\mathcal{D}_{obs} \cup \mathcal{D}_{int}|$,

$$d_{\mathrm{TV}}(\hat{p}, p^\star) \leq \varepsilon, \text{ where } \quad \varepsilon \to 0 \quad \text{as } n \to \infty.$$

**(A2)** (**Bounded log-likelihood ratios**). There exists $B < \infty$ such that for all $(\mathcal{G}, \mathbf{R})$,

$$\left| \log \frac{p(\mathbf{X}|\mathcal{G}^\star,\mathbf{R}^\star)}{p(\mathbf{X}|\mathcal{G},\mathbf{R})} \right| \leq B.$$

(A1) guarantees $\hat{p}$ is close to $p$ with high probability as the sample size grows. (A2) keeps randomness well-behaved so we can get sharp concentration. With Lemma 4.1, we show that the posterior converges in $p^\star$ to 1 as the sample size increases via Theorem 4.3. Theorem 4.4 says that the posterior is at least one minus an error term that decays exponentially in $n$. The error term scales with the number of wrong root cause candidates $M$, the gap between the ideal KL-divergence gap $\Delta_{\min}$ and the cost $2B\varepsilon$ we pay for using $\varepsilon$-accurate plug-in distribution.

**Theorem 4.3** (Posterior consistency for multiple root causes with $\varepsilon$-vanishing approximation). *Let $\mathbf{R}^\star$ be a set of root causes. Under (A1)–(A2) and causal sufficiency, we have*

$$p(\mathcal{G}^\star, \mathbf{R}^\star \mid \mathcal{D}) \xrightarrow[n \to \infty]{p^\star} 1. \quad (6)$$

**Theorem 4.4** (Finite-sample bound with $\varepsilon$-robustness). *For any $\delta \in (0, 1)$, let $M$ be the number of wrong pairs $(\mathcal{G}, \mathbf{R})$ considered and let $\Delta^{\mathrm{eff}}_{\min}(n) := \Delta_{\min} - 2B\varepsilon$, $t_n := B\sqrt{\frac{2\ln(2M/\delta)}{n}}$. With probability at least $1 - \delta - \eta$,*

$$p(\mathcal{G}^\star, \mathbf{R}^\star \mid \mathcal{D}) \geq 1 - M \exp\left\{-n\left(\Delta^{\mathrm{eff}}_{\min}(n) - t_n\right)\right\}$$
$$\times \max_{(\mathcal{G},\mathbf{R}) \neq (\mathcal{G}^\star,\mathbf{R}^\star)} \frac{p(\mathcal{G}, \mathbf{R})}{p(\mathcal{G}^\star, \mathbf{R}^\star)}. \quad (7)$$

**Uncertainty of CPDAGs** We now discuss how to handle the uncertainty of CPDAGs. Let $C$ be a random variable that takes different CPDAGs. We can extend our approach by estimating the posterior over CPDAGs $p(C|\mathcal{D})$. We can rewrite equation (3) as follows.

$$p(\mathbf{R}|\mathcal{D}) = \sum_C p(\mathbf{R}|C, \mathcal{D})P(C|\mathcal{D}), \quad (8)$$

$$p(\mathbf{R}|C, \mathcal{D}) = \frac{p(\mathcal{D}|\mathbf{R}, C)p(\mathbf{R}|C)}{\sum_{\mathbf{R}'} p(\mathcal{D}|\mathbf{R}', C)p(\mathbf{R}'|C)}, \quad (9)$$

$$p(\mathcal{D}|\mathbf{R}, C) = \sum_{\mathcal{G} \in C} p(\mathcal{D}|\mathcal{G}, \mathbf{R})p(\mathcal{G}|C, \mathbf{R}) \quad (10)$$

Note that we can evaluate $p(\mathbf{R}|C, \mathcal{D})$ in the same manner via Algorithm 1 for computing $p(\mathcal{D}|\mathcal{G}, \mathbf{R})$ by fixing one CPDAG at a time. The posterior over CPDAGs $p(C|\mathcal{D})$

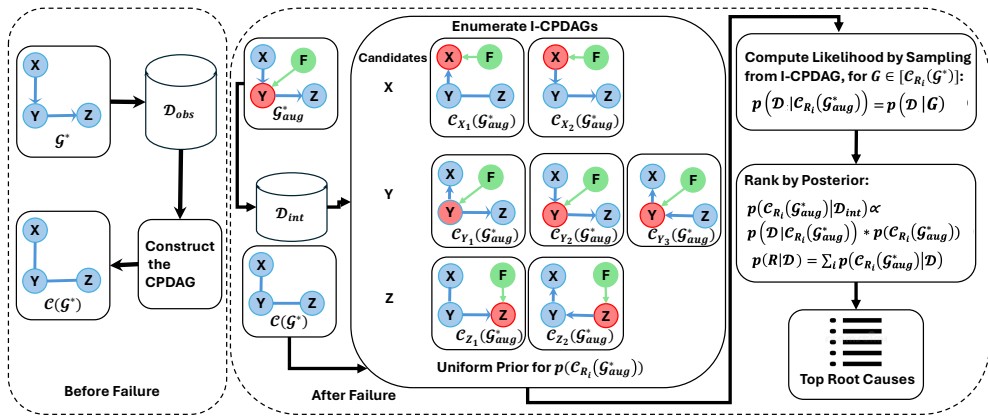

*Figure 1.* An illustration on the **BRCD**'s workflow. Consider the ground truth graph $\mathcal{G}^\star = [X \rightarrow Y \rightarrow Z]$ of a system, from which we collect the observational data $\mathcal{D}_{obs}$ before the failure happens. We recover the CPDAG $\mathcal{C}(\mathcal{G}^\star) = [X - Y - Z]$ using $\mathcal{D}_{obs}$. We use a variable $F$ (green) to represent the binary indicator that points to the root cause $Y$ (red), to construct the augmented graph $\mathcal{G}^\star_{aug} = [X \rightarrow Y \rightarrow Z, F \rightarrow Y]$. The interventional data $\mathcal{D}_{int}$ is collected after failure on $Y$. We list all the possible $\mathcal{I}$-CPDAGs by enumerating all potential root causes as $F$'s child and valid edge configurations of $F$'s child. Take $X$ for example, there are two possible $\mathcal{I}$-CPDAGs, $[F \rightarrow X \rightarrow Y \rightarrow Z]$ and $[F \rightarrow X \leftarrow Y - Z]$. For each candidate $R$, we compute $p(\mathcal{G}|R) = [[2/9, 1/9], [1/9, 1/9, 1/9], [2/9, 1/9]]$ as there are 9 possible augmented graphs in total ($X \rightarrow Y \leftarrow Z$ is excluded since $X - Y - Z$ is the given CPDAG) . For each candidate root cause $R$ and its possible $\mathcal{I}$-CPDAG, we randomly sample a DAG from the $\mathcal{I}$-CPDAG and estimate the likelihood according to the sampled DAG. As such, we can update the posteriors of each $\mathcal{I}$-CPDAG and aggregate them to get the posteriors for root causes. We output the top root causes according to the rank of the nodes' posteriors. With the first few anomaly samples, we can observe that the posterior mildly favors $R = Y$. As the number of anomaly samples grows, the posterior of $R = Y$ will get closer to $1$.

can be computed prior to failure for RCA. We can bootstrap samples multiple times and apply observational causal discovery on each bootstrapped sample to obtain a set of CPDAG candidates $C_b$, where $b \in [B]$ and $B$ is the number of candidates. For each candidate $C_b$, we compute the likelihood $p(\mathcal{D}|C_b)$ by sampling a single DAG from the corresponding CPDAG. We can assume a uniform prior $p(C_b)$ on these candidates and compute the unnormalized weight $w_b = p(\mathcal{D}|C_b)p(C_b)/q(C_b)$, where $q(C_b)$ is the empirical frequency of getting the CPDAG $C_b$ from the discovery algorithm. We divide by $q(C_b)$ to correct the fact that some CPDAGs appear more often simply because the bootstrap process generates them often, not because they truly have a higher posterior probability. Finally, we approximate the posterior via $p(C_b|\mathcal{D}) = w_b / \sum_b w_b$. Liu et al. (2025) have also proposed an algorithm named CPDAG-DFN using GFlowNet for estimating the posterior of CPDAGs.

So far we have discussed how to sample $\mathcal{G} \in [\mathcal{G}^\star]$ using an efficient CPDAG sampler to estimate $p(\mathcal{D}|\mathcal{G}, \mathbf{R})$. This can help decrease the number of posteriors we need to keep track of since we just need $p(\mathbf{R}|\mathcal{D})$. For an accurate update, one can consider $(\mathcal{G}, \mathbf{R})$ as a joint event. Next, we show that we just need to consider the $\mathcal{I}$-CPDAGs with $\mathbf{R}$ as the targets instead of the full MEC.

**Lemma 4.5.** *Let $\mathcal{G}$ be a DAG and $\mathbf{R}$ be a set of root cause candidates. Consider $(\mathcal{G}, \mathbf{R})$ as a joint event. With a uniform prior on $(\mathcal{G}, \mathbf{R})$, ranking the posteriors of $(\mathcal{G}, \mathbf{R})$ is*

*the same as ranking $\mathcal{I}$-CPDAGs.*

Lemma 4.5 shows that instead of tracking all $(\mathcal{G}, \mathbf{R})$ pairs, we just need to keep track of all the $\mathcal{I}$-CPDAGs with $\mathbf{R}$ as the target and different cut configurations of $\mathbf{R}$. This implies that our algorithm simultaneously discovers more causal relationships on the given partial causal structure.

## 5. Evaluation

### 5.1. Experiment Settings

**Synthetic dataset** We randomly generate 100 different DAGs of size $n \in \{10, 25, 50, 75, 100, 1000\}$ via PyAgrum (Ducamp et al., 2020) in Python. For each DAG, we generate 10000 observational samples and $10n$ anomalous samples. The number of edges is $1.5n$. We randomly pick one variable to alter its observational distribution to create distribution shifts as the anomaly. We limit the runtime to 3 minutes. We provide the standard errors based on 100 different DAGs for each $n$. The result is shown by Figure 2. We also conduct another experiment on the convergence of the interventional sample. We fix a DAG of size 50 with 75 edges. Then, we generate 10000 observational samples along with $m$ interventional samples, where $m \in \{5, 10, 100, 500, 1000\}$ by randomly picking a node $X$ and changing its conditional distribution $p(X \mid Pa(X))$. We provide the result in the Figure 3.

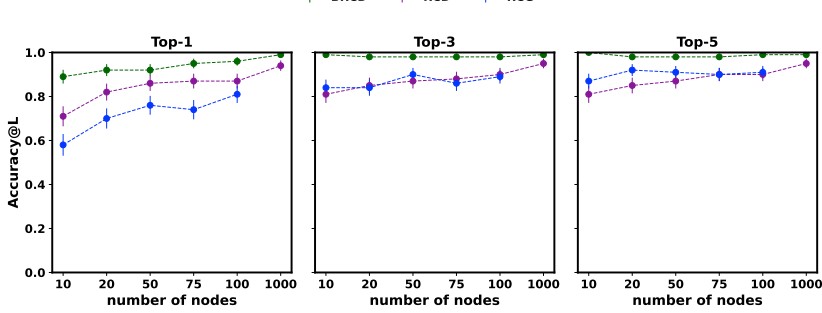 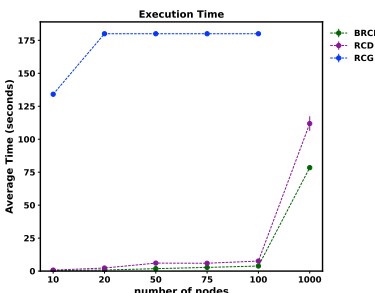

(a) Top-*l* accuracy of **BRCD** with baselines with discrete variables up to 4 states    (b) Execution time in failure period

*Figure 2.* The results demonstrate that **BRCD** (**green**) consistently achieves higher accuracy than baselines. It is followed by **RCD** (**purple**) and **RCG** (**blue**). The execution time of **BRCD** is close to that of **RCD**. **RCG** also takes a long time to compute conditional mutual information during the failure period. This experiment includes baselines that handle categorical variables.

| | | RCD | RCG | BARO | BRCD | BRCD-C | BRCD-M | BRCD-B10 | BRCD-B100 | SO | ST | Cholesky | IDI | ShapleyIQ | MicroDig | SimpleRCA |
|---|---|---|---|---|---|---|---|---|---|---|---|---|---|---|---|---|
| | high_traffic | 0.00 | 0.00 | 0.15 | 0.27 | **0.35** | 0.08 | 0.31 | **0.31** | 0.00 | 0.00 | 0.00 | 0.12 | 0.08 | 0.00 | 0.08 |
| | low_traffic | 0.31 | 0.15 | 0.27 | **0.46** | 0.42 | 0.35 | 0.42 | **0.46** | 0.04 | 0.00 | 0.00 | 0.12 | 0.15 | 0.00 | 0.15 |
| **top-1** | temporal_traffic1 | 0.25 | 0.12 | 0.25 | 0.38 | **0.63** | **0.63** | 0.25 | 0.50 | 0.00 | 0.00 | 0.00 | 0.25 | 0.12 | 0.00 | 0.25 |
| | temporal_traffic2 | 0.62 | 0.25 | 0.25 | 0.50 | **0.88** | **0.88** | 0.62 | 0.62 | 0.00 | 0.00 | 0.00 | 0.25 | 0.25 | 0.00 | 0.25 |
| | Average | 0.30 | 0.13 | 0.23 | 0.40 | **0.57** | 0.48 | 0.40 | 0.47 | 0.01 | 0.00 | 0.00 | 0.19 | 0.15 | 0.00 | 0.18 |
| | high_traffic | 0.00 | 0.00 | 0.23 | 0.35 | **0.38** | 0.23 | 0.31 | 0.35 | 0.00 | 0.00 | 0.00 | 0.23 | 0.23 | 0.42 | 0.19 |
| | low_traffic | 0.38 | 0.19 | 0.42 | 0.65 | 0.69 | 0.58 | **0.77** | 0.73 | 0.04 | 0.04 | 0.00 | 0.23 | 0.38 | 0.62 | 0.31 |
| **top-3** | temporal_traffic1 | **0.75** | 0.25 | 0.38 | 0.62 | 0.75 | 0.75 | 0.62 | **0.75** | 0.12 | 0.00 | 0.00 | 0.38 | 0.38 | 0.62 | 0.38 |
| | temporal_traffic2 | 0.75 | 0.25 | 0.38 | 0.62 | **0.88** | **0.88** | **0.88** | **0.88** | 0.00 | 0.00 | 0.00 | 0.50 | 0.50 | 0.62 | 0.38 |
| | Average | 0.47 | 0.17 | 0.35 | 0.56 | **0.68** | 0.61 | 0.65 | **0.68** | 0.04 | 0.01 | 0.00 | 0.34 | 0.37 | 0.57 | 0.32 |
| | high_traffic | 0.00 | 0.23 | 0.31 | 0.38 | **0.42** | 0.38 | 0.35 | **0.42** | 0.00 | 0.00 | 0.00 | 0.35 | 0.31 | 0.62 | 0.35 |
| | low_traffic | 0.42 | 0.31 | 0.46 | 0.73 | 0.73 | 0.58 | 0.81 | **0.85** | 0.08 | 0.23 | 0.04 | 0.35 | 0.46 | 0.65 | 0.46 |
| **top-5** | temporal_traffic1 | **0.75** | 0.38 | 0.38 | 0.62 | **0.75** | **0.75** | 0.62 | **0.75** | 0.25 | 0.00 | 0.00 | 0.38 | 0.38 | 0.62 | 0.38 |
| | temporal_traffic2 | 0.75 | 0.38 | 0.50 | 0.75 | **0.88** | **0.88** | **0.88** | **0.88** | 0.00 | 0.00 | 0.12 | 0.62 | 0.62 | 0.62 | 0.38 |
| | Average | 0.48 | 0.33 | 0.41 | 0.62 | 0.69 | 0.65 | 0.67 | **0.73** | 0.08 | 0.06 | 0.04 | 0.43 | 0.44 | 0.63 | 0.39 |

*Table 1.* **Results for the Petshop dataset** (Hardt et al., 2023). Scenario-level pooled top-*l* accuracy of the baselines, the proposed algorithm **BRCD** and its bootstrapping variants: **BRCD-B10** and **BRCD-B100**, where **BRCD-B10** and **BRCD-B100** use 10 and 100 bootstrapped observational samples, respectively. We also include two additional **BRCD** variants: **BRCD-C** and **BRCD-M**, where **BRCD-C** directly uses the service map in Petshop data without using any CPDAG from causal discovery algorithms and **BRCD-M** enforces the ancestral relations from the service map in the causal discovery algorithm named BOSS used by **BRCD**. Each cell is a case-weighted aggregate over two cases named *availability* and *latency*. There are a total of 26, 26, 8, and 8 datasets for high_traffic, low_traffic, temporal_traffic1, and temporal_traffic2, respectively. Each dataset contains only 5 anomalous samples.

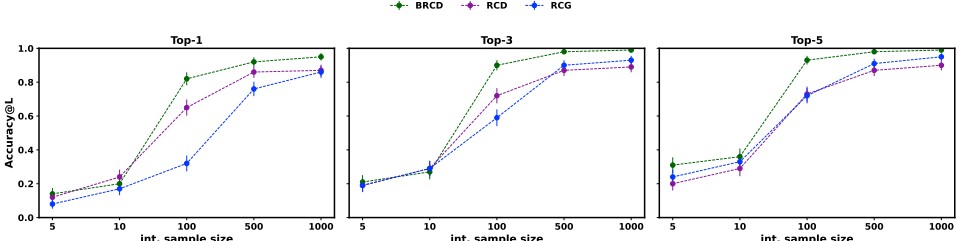

*Figure 3.* Top-*l* accuracy with 10,000 observational samples and increasing interventional samples for 50 nodes.

Hardt et al. (2023) introduced the **Petshop** dataset to benchmark root cause analysis (RCA) methods in microservice systems. Collected from a real microservice application, the dataset contains 68 injected performance faults that propagate through the system, increasing end-to-end latency and degrading service availability. Here, latency and availability are application *metrics*: latency measures the time required to process a request, while availability measures the proportion of requests that return an error. The dataset further includes three traffic scenarios: high, low, and temporal, capturing different load levels and periodic demand patterns. The dataset contains roughly 40 continuous variables after data preprocessing. Petshop poses a difficult RCA setting due to near-constant variables and substantial missing values.

For high and low traffic, there are roughly 590 observational samples. For temporal traffic, there are 1656 observational samples. All scenarios have only 5 anomalous samples. The data comes with a service map, which is often utilized to create a proxy causal graph. The result is shown in Table 1. We provide additional real-world experiments on Online Boutique (OB) and Sockshop applications in Appendix G.

**Evaluation Metrics** For the synthetic and the Petshop experiments, top-$l$ accuracy refers to the frequency of getting the true root cause in the $l$-th rank output by the algorithm. We also report the performance in terms of mean reciprocal rank in Table 7 in the Appendix.

**Algorithms** We compare with the following state-of-the-art RCA methods: **BARO** (Pham et al., 2024), **RCD** (Ikram et al., 2022), **RCG** (Ikram et al., 2025) , **IDI** (Nagalapatti et al., 2025), **Cholesky** (Li et al., 2025), **SCORE TRAVERSAL (ST)**, **SCORE ORDERING (SO)** (Orchard et al., 2025), **ShapleyIQ** (Li et al., 2024), **MicroDig** (Tao et al., 2024) and **SimpleRCA** (Fang et al., 2025). For the brief description of these algorithms, please see Appendix C. We give four variants of **BRCD**: **BRCD-C**, **BRCD-M**, **BRCD-B10** and **BRCD-B100**, where **BRCD-C** takes a service map instead of the CPDAG given by a causal discovery algorithm, **BRCD-M** enforces ancestral relations in the CPDAG learning prcoess of the discovery algorithm named BOSS (Andrews et al., 2023), **BRCD-B10** and **BRCD-B100** use 10 and 100 bootstrapped observational samples respectively. **BRCD** uses a ground truth CPDAG in the synthetic experiment to demonstrate the merits of our theoretical results about posterior consistency and our efficient graph sampling method. For other experiments, **BRCD** takes a CPDAG output by an observational causal discovery algorithm named BOSS (Andrews et al., 2023). Note that exploiting a partial causal structure is not a plug-and-play capability for most RCA algorithms (Ikram et al., 2025). Please see Appendix D for more implementation details.

### 5.2. Performance Assessment

**Accuracy** In the synthetic experiment, as shown by Figure 2a, we can see that **BRCD** consistently achieves the highest Top-$l$ accuracies across graph sizes. The accuracy converges to 1 with relatively few samples, supporting our Bayesian learning design. In the Petshop experiment, **BRCD** outperforms all baselines in top-1 accuracy (Table 1), demonstrating **BRCD**'s sample efficiency for RCA. We see that the boostrapping can further improve performance. When expert knowledge is provided to **BRCD**, we also see that there is improvement as shown by the performance of **BRCD-C**. In the OB and Sockshop experiments, **BRCD** slightly underperforms **SimpleRCA** and **BARO** while it outperforms other baselines. We note that **BARO** and **Sim-**

pleRCA use a similar strategy to rank root causes by measuring the deviation of the marginal distributions between the anomalous and normal data. While this strategy performs well on Sockshop and OB datasets, it does not generalize well to the Petshop data.

**Scalability** From Figure 2b, **BRCD** scales to 1000 nodes more efficiently than **RCD** and **RCG**. This highlights the effectiveness of our likelihood evaluation for posterior updates after failures occur, since other computational costs can be amortized during normal operation. **RCG** does not finish within the time limit.

**Interventional sample convergence** Based on Figure 3, **BRCD** converges quickly in top-$l$ accuracy when identifying the true root cause. The advantage is most pronounced at an interventional sample size of 100, further supporting the posterior consistency of **BRCD**. Also, the finite-sample bound in Theorem 4.4 shows that the posterior on the true root cause converges to 1 at an exponential rate in the number of samples, governed by the effective gap $\delta_{eff} = \delta_{\min} - 2B\epsilon$. Empirically, this behavior is reflected in Figure 3. This aligns with the theory's prediction of fast concentration when the effective gap is positive. Similarly, the robustness under very limited anomalous samples (e.g. 5 samples in Petshop) is consistent with the bounds dependence on the KL separation rather than large sample sizes. We note that the bound is primarily qualitative rather than tight in practice, as it depends on unknown quantities such as and the approximation error. Directly estimating these terms is challenging, which limits precise numerical validation. However, the observed convergence trends are consistent with the theoretical guarantees.

**Real-world applicability** Across OB, Sockshop, and Petshop (Tables 1, 4, 5), **BRCD** achieves near-best performance on the OB and Sockshop datasets and the best performance on the Petshop dataset, even when the ground-truth CPDAG is unavailable. It leverages recent advances in observational causal discovery and an efficient graph-sampling scheme to enable scalable, sample-efficient RCA. In contrast to **IDI** and **ST**, **BRCD** does not require the trigger point a priori, nor full causal knowledge of all observed variables. Notably, **BRCD** remains competitive under extremely scarce failure data in Petshop, consistent with our Bayesian posterior update design. This matches real microservice incidents, where failures are rare and yield only a brief anomalous window, so RCA must work with very few abnormal samples. Also, the variants of **BRCD** can take a service map in the form of a DAG directly or incorporate it as background knowledge into its CPDAG learning process. We give a flexible framework to incorporate observational causal discovery, as there can be uncertainty about the causal relations at the metric-level, even if there exists a call graph at the

| Algorithm | Accuracy (mean ± stderr) |
|---|---|
| RCG | $0.80 \pm 0.04$ |
| **BRCD** | $\mathbf{0.90 \pm 0.03}$ |
| **RCD** | $\mathbf{0.90 \pm 0.03}$ |

*Table 2.* Results for different algorithms that can handle categorical data for detecting two root causes that are randomly chosen in 100 different DAGs. The accuracy is measured by counting how many times the algorithm can capture the two root causes as the first two nodes in the output.

| Metric | BRCD | RCD | RCG |
|---|---|---|---|
| Top-1 | $\mathbf{0.60 \pm 0.07}$ | $0.50 \pm 0.07$ | $0.36 \pm 0.06$ |
| Top-3 | $\mathbf{0.80 \pm 0.06}$ | $0.64 \pm 0.07$ | $0.60 \pm 0.06$ |
| Top-5 | $\mathbf{0.90 \pm 0.04}$ | $0.74 \pm 0.06$ | $0.74 \pm 0.06$ |

*Table 3.* Performance comparison on the same algorithms in the synthetic experiment without ground truth CPDAG.

service level. We see that both **BRCD-C** and **BRCD-B100** improve on **BRCD** and outperform other baselines on the Petshop data.

### 5.3. Additional Experiments

Next, we further examine **BRCD** under multiple root causes and in the presence of latent confounders by extending the synthetic experiments.

**Multiple Root Causes** We conduct an experiment that generate 100 random DAGs of size 50 with 10,000 observational samples and 5000 interventional samples using Python `pyagrum` library (Ducamp et al., 2020). Each variable has up to 4 states randomly assigned. The number of edges is 75. We randomly perturb two nodes in the DAG as a soft intervention by altering the conditional probability table. We evaluate the algorithms by counting whether the first two outputs of the algorithm can capture the intervened nodes. We report the average accuracy in Table 2. Both **BRCD** and **RCD** have achieved the highest accuracy of 0.9. We observe that **BRCD** leverages the ground-truth CPDAG more effectively than **RCG** as computing conditional mutual information in **RCG** can be sample-inefficient.

**Presence of Latent Confounders** We have additionally conducted an experiment where we use a learned CPDAG in the presence of latent confounders to help understand the performance better without a ground-truth CPDAG to supplement our synthetic experiment. We follow a similar setup in our synthetic experiment. We generate 50 different DAGs per graph size 10, with an additional 5 more unobserved confounders added to the DAGs and $2n$ many edges. We choose a small graph size because scaling the learned CPDAG from discrete data is challenging, as we use the BOSS algorithm, which uses BIC for model selection. Switching to discrete BDeu scores does not scale equally fast. The root cause is randomly picked among the observed variables. Each node has 4 states. We generate 10,000 observational samples with 100 anomalous samples. Here, **BRCD** uses the learned CPDAG based on the BeDu score, based on the BOSS causal discovery algorithm from the observed data. **RCG** also uses the 0-order essential graph. We report the result in terms of top-$l$ accuracy in Table 3.

We see that **BRCD** still outperforms **RCD** and **RCG** in the presence of latent confounders. One reason is that our Bayesian method does not rely on conditional independence (CI) tests like **RCD** to determine root causes.

## 6. Discussions

We presented the first Bayesian root cause discovery algorithm with a statistical consistency guarantee given a partial causal structure. Our algorithm is not restricted to any parametric assumption. It also does not impose any functional relationship on the underlying causal structure among system metrics. Another favorable property of our proposed method is its *anytime* nature. Specifically, Bayesian updates of the root cause posteriors can be performed at any stage as new data becomes available. Since the method only requires maintaining the posterior distributions over the $n$ candidate root causes, the update process remains computationally tractable and efficient throughout.

One limitation of our algorithm is its reliance on the partial causal structure, which may not be available at hand. However, we argue that a partial causal structure can be obtained in the pre-failure time via advanced causal discovery algorithms. We use a polynomial-time DAG sampling procedure leveraging this partial structure. We have also shown how to deal with the uncertainty of CPDAGs in our proposed method. Furthermore, we provide the identifiability result of root causes under this sampling procedure to localize the root cause via posterior approximation. The overall computational efficiency of our algorithm depends on the choice of the estimator for approximating the posterior, the number of variables, and the degree of the causal graph. Another limitation is the estimation of the joint distribution given the potential root causes and the sampled DAG with limited anomalous samples. We provide practical guidance on estimating likelihood in Appendix F. Besides, we show that our root cause ranking procedure simultaneously ranks the possible $\mathcal{I}$-CPDAGs. Moreover, **BRCD** outperforms the state-of-the-art baselines based on the overall performance across synthetic and real-world experiments. For future work, we want to develop a tighter bound on the sample complexity of the algorithm. It would also be interesting to explore how the bound ties to various priors for **BRCD** based on other RCA techniques or expert knowledge.

## Acknowledgements

This research has been supported in part by NSF CAREER 2239375, IIS 2348717, Amazon Research Award, Adobe Research and Intuit.

## Impact Statement

The proposed Bayesian root cause analysis algorithm has the potential for significant positive societal impact by enhancing the reliability and maintainability of complex systems such as cloud-native applications and microservice architectures. By enabling the efficient identification of root causes of system failures, the method can reduce downtime, improve user experiences, and lower operational costs for organizations. This could benefit industries ranging from healthcare to finance, where system failures may have high-stakes consequences. Moreover, the approach may help democratize troubleshooting by reducing reliance on deep domain expertise. However, the deployment of automated root cause analysis also carries potential risks. Misidentification of root causes, especially in high-stakes systems, could lead to misguided interventions, compounding failures or introducing new vulnerabilities. Additionally, as system observability and intervention capabilities expand, there may be privacy concerns depending on how diagnostic data is collected and used. Careful evaluation and responsible deployment practices will be essential to ensure the method's benefits are realized without exacerbating risks. The same posterior-ranking perspective may also inform biological diagnostics, such as marker-gene prioritization in interpretable scRNA-seq analysis (Zhou et al., 2022; Lu et al., 2021; Plumb et al., 2020), though such applications would require domain-specific validation.

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

# Appendix

## A. Related Work

**Causality-based Root Cause Analysis in Microservices.** Prior studies in causal discovery for RCA in microservices mostly focus on improving the accuracy and time efficiency of learning a causal graph about the monitoring data in the post-failure period. A plethora of works are based on a well-celebrated causal discovery algorithm known as PC algorithm (Spirtes et al., 2001) and its variants to construct a causal graph (Chen et al., 2014; Wang et al., 2018; Lin et al., 2018; Ma et al., 2019; Chen et al., 2019; Meng et al., 2020; Ma et al., 2020). Ikram et al. (2022) incorporates a divide-and-conquer strategy to split all the metrics into smaller subgraphs and identify the root cause without learning a complete causal structure from these subgraphs. A common issue with these methods is their reliance on conditional independence tests. As the conditional set size increases, the statistical power of the test also quickly diminishes (Shah & Peters, 2020). Small sample sizes only further exacerbate this issue, making it challenging for these methods to perform effectively in scenarios where only limited anomalous samples are available. While others have explored constructing causal graphs using methods such as DirectLiNGAM (Wu et al., 2021), variational autoencoder (Xin et al., 2023), neural Granger causal discovery with contrastive learning (Lin et al., 2024), using textual description of the metrics via large language models (Xie et al., 2024), the sample efficiency of these approaches remains unclear. Our Bayesian approach also offers a principled uncertainty quantification over the existing scoring methods that require a DAG such as random walk (Wang et al., 2018; Ma et al., 2020), PageRank (Wu et al., 2021; Xin et al., 2023; Lin et al., 2024), BFS (Lin et al., 2018), DFS (Chen et al., 2014). In contrast to parametric approaches that depend on restrictive assumptions of linearity and non-Gaussianity (Strobl & Lasko, 2023), our method adopts a non-parametric RCA framework (Orchard et al., 2025). Furthermore, it relaxes the requirement of a fully specified causal graph, unlike prior methods (Budhathoki et al., 2022; Nagalapatti et al., 2025). Similar to Ikram et al. (2025), our method builds on a CPDAG, which can be learned under less restrictive assumptions in causal discovery.

**Observational Causal Discovery.** Most of the existing work focuses on learning causal relations between the metrics after a failure occurs. However, it is possible to start constructing the causal graph when the monitoring data is available before the failure through observational causal discovery. Generally, one can learn up to an MEC of DAGs without further assumptions. Many causal discovery algorithms can learn a MEC in the form of CPDAG from observational data such as PC (Spirtes et al., 2001), GaRsP (Lam et al., 2022), GES and its variants (Chickering, 2002; 2020; Nazaret & Blei, 2024). Recent research has demonstrated the ability to learn such partial causal structures in a scalable and accurate manner (Andrews et al., 2023), which aligns well with the requirements of RCA in microservices involving complex networks. Recent scalable causal discovery methods have also explored divide-and-conquer strategies, such as learning local Markov-blanket subgraphs and integrating them into a global causal structure, using distributed subproblem reconciliation, causal graph partitioning, or voting-based subgraph integration (Dong et al., 2025; Shah et al., 2024). These approaches are complementary to BRCD, which uses a partial pre-failure causal structure to perform Bayesian root-cause inference rather than to solve the causal discovery problem itself.

**Causal Discovery with Unknown Interventions.** Given the result from Ikram et al. (2022) that models failures in a system as an intervention, some causal discovery literature that can potentially be applied for RCA in microservices (Mooij et al., 2020; Kocaoglu et al., 2019; Squires et al., 2020; Jaber et al., 2020; Wang et al., 2022). However, they all rely on conditional independence tests. As noted earlier, these hypothesis tests often demand large sample sizes, rendering the methods less robust for realistic datasets with limited samples. In contrast, Bayesian approaches give principled uncertainty quantification. This approach is advantageous given the substantial epistemic uncertainty in real-world systems like microservices, where anomalous samples are scarce. In Bayesian causal discovery with unknown interventions, Eaton & Murphy (2007) is constrained to discrete variables and utilizes a dynamic programming approach that lacks scalability for large graphs. Hägele et al. (2023) formulates continuous relaxations of the joint inference problem under unknown interventions, learning both the causal graph and the interventional targets at the same time. However, their approach assumes the differentiability of the data-generating process to leverage gradient-based optimization methods. Also, our approach shows that jointly learning the true causal graph is not necessary for learning the interventional target given a partial causal structure in Bayesian causal discovery. Linearity has also been exploited for discovering interventional targets. Taeb et al. (2023) developed a regularized maximum-likelihood estimator for identifying unknown interventional targets in a Gaussian model with latent variables. Rothenhäusler et al. (2015) proposed an algorithm named BACKSHIFT to identify unknown interventional targets under linear models with cycles. Varici et al. (2021) constructed a scalable algorithm to find unknown targets without full causal structure learning by leveraging sparse changes in precision matrix differences between datasets.

# B. Proofs

## B.1. Proof of Lemma 4.1

**Lemma B.1** (Generic Identifiability of multiple root causes). *Let $\mathcal{G} = (\mathbf{V}, \mathbf{E})$ be a DAG and let $\mathbf{R} \subseteq \mathbf{V}$ be a set of root causes. Under* modularity, *positivity (all relevant conditionals are strictly positive on their supports), and that the nodewise mechanisms are* non-degenerate *(for $V \in \mathbf{R}$ there exist parent values and two child values with different conditional likelihood ratios across F), then for almost all parameter values (excluding a measure-zero subset), any two distinct target sets $\mathbf{R} \neq \mathbf{R}'$ induce distinct interventional families $\{p(\mathbf{X} \mid F = f, \mathcal{G})\}_{f \in \{0,1\}}$. In particular, if $(\mathcal{G}^\star, \mathbf{R}^\star)$ is the ground truth, then*

$$\Delta_{\min} := \inf_{(\mathcal{G}, \mathbf{R}) \neq (\mathcal{G}^\star, \mathbf{R}^\star)} \mathbb{E}_{p^\star} \left[ \log \frac{p(\mathbf{X} \mid \mathcal{G}^\star, \mathbf{R}^\star)}{p(\mathbf{X} \mid \mathcal{G}, \mathbf{R})} \right] > 0. \tag{5}$$

*Proof.* By interventional faithfulness and causal sufficiency, adding the edge $F \to R^\star$ yields an I-CPDAG in which all wrong $(G, R)$ induce distributions that are not Markov equivalent to $(G^\star, R^\star)$; hence their expected log-likelihood differs by a positive margin.

By modularity, across regimes $F = \{0, 1\}$, the only conditionals that may change are those for $V \in \mathbf{R}$:

$$p(\mathbf{X} \mid F = f, \mathcal{G}) = \prod_{V \in \mathbf{V}} p^{(f \cdot \mathbf{1}\{V \in \mathbf{R}\})}\big(V \mid Pa_{\mathcal{G}}(V)\big).$$

Then, we can define

$$L_{\mathbf{R}}(\mathbf{X}) := \log \frac{p(\mathbf{X} \mid F = 1, \mathcal{G})}{p(\mathbf{X} \mid F = 0, \mathcal{G})} = \sum_{V \in \mathbf{R}} \ell_V(\mathbf{X}), \qquad \ell_V(\mathbf{X}) := \log \frac{p^{(1)}(V \mid Pa_{\mathcal{G}}(V))}{p^{(0)}(V \mid Pa_{\mathcal{G}}(V))}. \tag{11}$$

By construction, each $\ell_V$ depends only on $(V, Pa_{\mathcal{G}}(V))$ and on the local parameters governing node $V$.
**Claim.** If $\mathbf{R} \neq \mathbf{R}'$, then *generically* $L_{\mathbf{R}} \not\equiv L_{\mathbf{R}'}$ as functions of $\mathbf{X}$. Hence, the conditional families differ.

*Proof of claim by contradiction.* Suppose $\mathbf{R} \neq \mathbf{R}'$ but $L_{\mathbf{R}}(\mathbf{X}) \equiv L_{\mathbf{R}'}(\mathbf{X})$ for all $\mathbf{X}$ for a given choice of parameters. Let $\mathbf{S} := \mathbf{R} \triangle \mathbf{R}'$ (symmetric difference) and pick a node $W \in \mathbf{S}$ that has no descendants in $\mathbf{S}$. Without loss of generality, $W \in \mathbf{R} \setminus \mathbf{R}'$.

Rewrite the identity $L_{\mathbf{R}} \equiv L_{\mathbf{R}'}$ as

$$D(\mathbf{X}) := \sum_{V \in \mathbf{R} \setminus \mathbf{R}'} \ell_V(\mathbf{X}) - \sum_{U \in \mathbf{R}' \setminus \mathbf{R}} \ell_U(\mathbf{X}) \equiv 0. \tag{12}$$

Since $W$ has no descendants in $\mathbf{S}$, $W \notin Pa_{\mathcal{G}}(U)$ for every $U \in \mathbf{S} \setminus \{w\}$. Therefore each $\ell_U$ with $U \neq W$ is *independent of* $W$. It depends only on $(U, Pa_{\mathcal{G}}(U))$ and does not involve $W$.

By non-degeneracy at $W$, there exist some fixed parent values $\mathbf{a} \in \mathcal{X}^{|Pa_{\mathcal{G}}(W)|}$ and two child values $b \neq b'$ in the support such that

$$\ell_W\big(W = b, \ Pa_{\mathcal{G}}(W) = \mathbf{a}, \mathbf{V} \setminus (W \cup Pa_{\mathcal{G}}(W)) = \mathbf{q}\big) \neq \ell_w\big(W = b', \ Pa_{\mathcal{G}}(W) = \mathbf{a}, \mathbf{V} \setminus (W \cup Pa_{\mathcal{G}}(W)) = \mathbf{q}\big) \tag{13}$$

Choose two assignments $\mathbf{x}, \mathbf{x}'$ that are identical except at $W$, where $(\mathbf{x}_w, \mathbf{x}'_w) = (b, b')$, and such that $\mathbf{x}_{Pa(W)} = \mathbf{x}'_{Pa(W)} = \mathbf{a}$; fix all other coordinates so that positivity holds. For these two assignments, all terms $\ell_U$ with $U \neq W$ take the *same* value (they do not depend on $W$), while $\ell_W$ differs. Consequently,

$$D(\mathbf{x}) - D(\mathbf{x}') = \ell_W(\mathbf{x}) - \ell_W(\mathbf{x}') \neq 0,$$

contradicting equation (12). Hence $L_{\mathbf{R}} \not\equiv L_{\mathbf{R}'}$. The only way the contradiction could be avoided is if $\ell_W$ were a.s. constant in the value $W$ takes. However, this violates the non-degeneracy assumption. To see it, let $c = \frac{p^{(1)}(W|\mathbf{a})}{p^{(0)}(W|\mathbf{a})}$. Then, integrate over $W$

$$1 = \int p^{(1)}(W \mid \mathbf{a}) dW = c \int p^{(0)}(W \mid \mathbf{a}) dW = c \tag{14}$$

Hence, $p^{(1)}(W \mid \mathbf{a}) = p^{(0)}(W \mid \mathbf{a})$, but non-degeneracy requires the node's mechanism to actually change across $F$ for some $\mathbf{a}$. The other way to avoid contradiction is that the parameters are so specially tuned that two different sums of local log-ratios coincide for every data vector e.g. $L_{\mathbf{R}} \equiv L_{\mathbf{R}'}$ for all $\mathbf{x}$. Then, for any fixed parent value $\mathbf{a}$ and two child values $b = b'$, we have $\ell_W(b; \mathbf{a}) = \ell_W(b'; \mathbf{a})$. That implies $\ell_W(\cdot; \mathbf{a}) = c_0$ for some constant $c_0$ on the support $W$. Then, following the same argument we made previously about $c = \frac{p^{(1)}(W|\mathbf{a})}{p^{(0)}(W|\mathbf{a})}$. We also have $p^{(1)}(\cdot \mid \mathbf{a}) = p^{(0)}(\cdot \mid \mathbf{a})$, leading to the same violation of the non-degeneracy assumption. Next, we want to explain why this set of parameters where $L_{\mathbf{R}} \equiv L_{\mathbf{R}'}$ can have measure zero. With discrete variables and positivity, each conditional $p^{(f)}(\cdot|\mathbf{a})$ lives in an open simplex. The constraints $p^{(1)}(\cdot \mid \mathbf{a}) = p^{(0)}(\cdot \mid \mathbf{a})$ are linear equalities inside that open set. They reduce the dimension by the number of child values $-1$ per $\mathbf{a}$, yielding a lower-dimensional subset. A finite intersection of lower-dimensional subsets inside a full-dimensional open set has Lebesgue measure zero. Therefore, the parameter set where the cross-node cancellation can hold for all $\mathbf{x}$ has measure zero. In other words, the only way two different set of root causes can produce $L_{\mathbf{R}} = L_{\mathbf{R}'}$ for all $\mathbf{x}$ is if every node in their symmetric difference has identical conditionals across the regime $f \in \{0, 1\}$. Therefore, the claim holds *generically*.

Lastly, we note that KL-divergence equals zero if and only if the two distributions are equal almost surely. Since we show that a wrong pair $(\mathcal{G}, \mathbf{R})$ induce a distribution $p(\mathbf{X}|\mathcal{G}, \mathbf{R})$ that is not equal to the true distribution $p(\mathbf{X}|\mathcal{G}^\star, \mathbf{R}^\star)$ generically, we have that $D_{KL}(p(\mathbf{X}|\mathcal{G}, \mathbf{R})||p(\mathbf{X}|\mathcal{G}^\star, \mathbf{R}^\star)) > 0$. $\Delta_{\min}$ is the infimum of those KL-divergence values over all wrong pairs. Since the candidate set of targets is finite, taking the minimum of finitely many strictly positive numbers is still strictly positive. Hence, we have $\Delta_{\min} > 0$. $\square$

### B.2. Proof of Corollary 4.2

*Proof.* For a target set $\mathbf{R}$ and the observational CPDAG $\mathcal{C}(\mathcal{G}^\star)$. Consider the loop in Algorithm 1 that ranges over all orientations of the edge cut $E[\mathbf{R}, \mathbf{V} \setminus \mathbf{R}]$ and then applies Meek rules to closure.

Take any orientation of $E[\mathbf{R}, \mathbf{V} \setminus \mathbf{R}]$ that is valid, i.e., it is extendable to at least one DAG $\mathcal{G}$ in the MEC represented by $\mathcal{C}(\mathcal{G}^\star)$. Starting from $\mathcal{C}(\mathcal{G}^\star)$, orienting the cut accordingly, and closing under Meek rules yields a maximally oriented CPDAG $\mathcal{C}$ that admits at least one DAG extension (namely $\mathcal{G}$) and contains all compelled orientations implied by the constraints. Hence $\mathcal{C}$ is an $\mathcal{I}$-CPDAG representing an $\mathcal{I}$-MEC compatible with $(\mathcal{C}(\mathcal{G}^\star), \mathbf{R})$.

Conversely, let $\mathcal{C}$ be any $\mathcal{I}$-CPDAG compatible with $(\mathcal{C}(\mathcal{G}^\star), \mathbf{R})$. Then $\mathcal{C}$ has a DAG extension $\mathcal{G}$ that is also consistent with $\mathcal{C}(\mathcal{G}^\star)$. Let $q_\mathcal{G}$ denote the orientation that $\mathcal{G}$ induces on the cut edges $E[\mathbf{R}, \mathcal{V} \setminus \mathbf{R}]$. Algorithm 1 considers $q_\mathcal{G}$ and, after applying Meek rules to closure, returns the maximally oriented CPDAG that represents the $\mathcal{I}$-MEC of $\mathcal{G}$, which is exactly $\mathcal{C}$. Therefore every compatible $\mathcal{I}$-CPDAG appears in the enumeration.

Combining both directions, Algorithm 1 enumerates all and only the $\mathcal{I}$-CPDAGs compatible with $(\mathcal{C}(\mathcal{G}^\star), \mathbf{R})$, equivalently all compatible $\mathcal{I}$-MECs. $\square$

### B.3. Proof of Theorem 4.3

**Theorem 4.3** (Posterior consistency for multiple root causes with $\varepsilon$-vanishing approximation). *Let $\mathbf{R}^\star$ be a set of root causes. Under (A1)–(A2) and causal sufficiency, we have*

$$p(\mathcal{G}^\star, \mathbf{R}^\star \mid \mathcal{D}) \xrightarrow[n \to \infty]{p^\star} 1. \tag{6}$$

**Lemma B.2** (Stability of the expected log-likelihood ratio under $\varepsilon$-perturbations). *Let $f_{\mathcal{G},\mathbf{R}}(\mathbf{x}) = \log \frac{p(\mathbf{x}|\mathcal{G}^\star, \mathbf{R}^\star)}{p(\mathbf{x}|\mathcal{G}, \mathbf{R})}$. Under assumption* (A2)*, for any $(\mathcal{G}, \mathbf{R})$ and any probability measures $Q, P$ on the sample space,*

$$\left| \mathbb{E}_Q[f_{\mathcal{G},\mathbf{R}}(\mathbf{X})] - \mathbb{E}_P[f_{\mathcal{G},\mathbf{R}}(\mathbf{X})] \right| \leq 2B \, d_{\mathrm{TV}}(Q, P). \tag{15}$$

*Consequently, on the event $\{d_{\mathrm{TV}}(\hat{p}, p^\star) \leq \varepsilon\}$,*

$$\mathbb{E}_{\hat{p}}[f_{\mathcal{G},\mathbf{R}}(\mathbf{X})] \geq \mathbb{E}_{p^\star}[f_{\mathcal{G},\mathbf{R}}(\mathbf{X})] - 2B\varepsilon. \tag{16}$$

*Proof.* By Assumption (A2), we have $|f_{\mathcal{G},\mathbf{R}}(\mathbf{x})| = |\log \frac{p(\mathbf{x}|\mathcal{G}^\star, \mathbf{R}^\star)}{p(\mathbf{x}|\mathcal{G}, \mathbf{R})}| \leq B$ for all $\mathbf{x}$. That implies the sup-norm $||f_{\mathcal{G},\mathbf{R}}||_\infty \leq B$. Let $g = f_{\mathcal{G},\mathbf{R}}/B$. Hence, $||g||_\infty \leq 1$. The variational definition of total variation says

$$d_{\mathrm{TV}}(Q, P) = \frac{1}{2} \sup_{||g||_\infty \leq 1} \left| \mathbb{E}_Q[g] - \mathbb{E}_P[g] \right|. \tag{17}$$

, which means $\big|\mathbb{E}_Q[g] - \mathbb{E}_P[g]\big| \leq 2d_{\mathrm{TV}}(Q,P)$. Then

$$\big|\mathbb{E}_Q f_{\mathcal{G},\mathbf{R}} - \mathbb{E}_P f_{\mathcal{G},\mathbf{R}}\big| = B\big|\mathbb{E}_Q[g] - \mathbb{E}_P[g]\big| \leq 2B\, d_{\mathrm{TV}}(Q,P).$$

Letting $(Q,P) = (\hat{p}, p^\star)$ and using Assumption (A1): $d_{\mathrm{TV}}(\hat{p}, p^\star) \leq \varepsilon$ yields the result. $\qquad\square$

*Proof of Theorem 4.3.* Let $\mathcal{M}$ be a finite collection of model indices, each of the form $(\mathcal{G}, \mathbf{R})$, where $\mathcal{G}$ is a causal graph and $\mathbf{R}$ is a set of root causes. For each $(\mathcal{G}, \mathbf{R}) \in \mathcal{M}$, let $p_{\mathcal{G},\mathbf{R}} = p(x|\mathcal{G}, \mathbf{R})$ denote the corresponding probability density. To simplify notations, we write $f := f_{\mathcal{G},\mathbf{R}} = \log\big(p_{\mathcal{G}^\star,\mathbf{R}^\star}/p_{\mathcal{G},\mathbf{R}}\big)$. By Assumption (A2), $f(\mathbf{X}_i) \in [-B, B]$. For any $t > 0$, Hoeffding's inequality yields

$$p\left(\frac{1}{n}\sum_{i=1}^{n} f(\mathbf{X}_i) < \mathbb{E}_{p^\star}[f(\mathbf{X})] - t\right) \leq \exp\left\{-\frac{2nt^2}{(2B)^2}\right\}. \tag{18}$$

By Lemma B.2, on the event $\{d_{\mathrm{TV}}(\hat{p}, p^\star) \leq \varepsilon_n\}$ we have

$$\mathbb{E}_{\hat{p}}[f] \geq \mathbb{E}_{p^\star}[f] - 2B\,\varepsilon \quad\Longleftrightarrow\quad \mathbb{E}_{p^\star}[f] \leq \mathbb{E}_{\hat{p}}[f] + 2B\,\varepsilon. \tag{19}$$

Combining (18) and (19), on $\{d_{\mathrm{TV}}(\hat{p}, p^\star) \leq \varepsilon\}$ we obtain

$$p\left(\frac{1}{n}\sum_{i=1}^{n} f(\mathbf{X}_i) < \mathbb{E}_{\hat{p}}[f] + 2B\,\varepsilon - t \;\Big|\; d_{\mathrm{TV}}(\hat{p}, p^\star) \leq \varepsilon\right) \leq \exp\left\{-\frac{2nt^2}{(2B)^2}\right\}. \tag{20}$$

Equivalently,

$$p\left(\frac{1}{n}\sum_{i=1}^{n} f(\mathbf{X}_i) \geq \mathbb{E}_{\hat{p}}[f] - \big(t - 2B\,\varepsilon_n\big) \;\Big|\; d_{\mathrm{TV}}(\hat{p}, p^\star) \leq \varepsilon\right) \geq 1 - \exp\left\{-\frac{2nt^2}{(2B)^2}\right\}. \tag{21}$$

Let

$$t_n := B\sqrt{\frac{2\ln\big(2M/\delta\big)}{n}}, \text{where } M = |\mathcal{H}| \qquad\text{and}\qquad s_n := t_n - 2B\,\varepsilon.$$

Since $\varepsilon \to 0$, we have $s_n > 0$ for all large $n$. Plugging $t = t_n$ into (21) gives, for each fixed $(\mathcal{G}, \mathbf{R}) \in \mathcal{H} := \mathcal{M}\backslash\{(\mathcal{G}^\star, \mathbf{R}^\star)\}$, for any $\delta \in (0,1)$

$$p\left(\frac{1}{n}\sum_{i=1}^{n} f_{\mathcal{G},\mathbf{R}}(X_i) \geq \mathbb{E}_{\hat{p}}[f] - s_n \;\Big|\; d_{\mathrm{TV}}(\hat{p}, p^\star) \leq \varepsilon\right) \geq 1 - \frac{\delta}{2M}. \tag{22}$$

Applying a union bound over $(\mathcal{G}, \mathbf{R}) \in \mathcal{H}$,

$$p\left(\bigcap_{(\mathcal{G},\mathbf{R})\in\mathcal{H}} \left\{\frac{1}{n}\sum_{i=1}^{n} f_{\mathcal{G},\mathbf{R}}(X_i) \geq \mathbb{E}_{\hat{p}}[f] - s_n\right\} \;\Big|\; d_{\mathrm{TV}}(\hat{p}, p^\star) \leq \varepsilon\right) \geq 1 - \frac{\delta}{2}.$$

Finally, combine with Assumption (A1), $p\big(d_{\mathrm{TV}}(\hat{p}, p^\star) \leq \varepsilon\big) \geq 1 - \eta$, to conclude

$$p(E_n) \geq 1 - \eta - \frac{\delta}{2}, \tag{23}$$

where

$$E_n = \left\{d_{\mathrm{TV}}(\hat{p}, p^\star) \leq \varepsilon\right\} \cap \left(\bigcap_{(\mathcal{G},\mathbf{R})\in\mathcal{H}} \left\{\frac{1}{n}\sum_{i=1}^{n} f_{\mathcal{G},\mathbf{R}}(X_i) \geq \mathbb{E}_{\hat{p}}[f_{\mathcal{G},\mathbf{R}}] - s_n\right\}\right).$$

On the event $E_n$ we have, for every wrong pair $(\mathcal{G}, \mathbf{R}) \in \mathcal{H}$,

$$\frac{1}{n}\sum_{i=1}^{n} f_{\mathcal{G},\mathbf{R}}(X_i) \geq \mathbb{E}_{\hat{p}}[f] - s_n. \tag{24}$$

Hence,

$$\sum_{i=1}^{n} f_{\mathcal{G},\mathbf{R}}(X_i) \geq n\Big(\mathbb{E}_{\hat{p}}[f] - s_n\Big). \tag{25}$$

For each wrong pair $(\mathcal{G}, \mathbf{R}) \in \mathcal{H}$, define

$$\Lambda_n(\mathcal{G}, \mathbf{R}) := \frac{p(\mathcal{G}, \mathbf{R} \mid \mathcal{D})}{p(\mathcal{G}^\star, \mathbf{R}^\star \mid \mathcal{D})} = \frac{p(\mathcal{G}, \mathbf{R})}{p(\mathcal{G}^\star, \mathbf{R}^\star)} \prod_{i=1}^{n} \frac{p(\mathbf{X}_i \mid \mathcal{G}, \mathbf{R})}{p(\mathbf{X}_i \mid \mathcal{G}^\star, \mathbf{R}^\star)}. \tag{26}$$

Then,

$$\Lambda_n(\mathcal{G}, \mathbf{R}) = \frac{p(\mathcal{G}, \mathbf{R})}{p(\mathcal{G}^\star, \mathbf{R}^\star)} \exp\Big\{-\sum_{i=1}^{n} f_{\mathcal{G},\mathbf{R}}(\mathbf{X}_i)\Big\}. \tag{27}$$

Equation (25) implies on $E_n$,

$$\Lambda_n(\mathcal{G}, \mathbf{R}) \leq \frac{p(\mathcal{G}, \mathbf{R})}{p(\mathcal{G}^\star, \mathbf{R}^\star)} \exp\Big\{-n\big(\mathbb{E}_{\hat{p}}[f] - s_n\big)\Big\}. \tag{28}$$

Next, we define

$$\Delta_{\min}^{\text{eff}}(n) := \Delta_{\min} - 2B\,\varepsilon, \tag{29}$$

where

$$\Delta_{\min} := \inf_{(\mathcal{G},\mathbf{R}) \neq (\mathcal{G}^\star, \mathbf{R}^\star)} \mathbb{E}_{p^\star}[f_{\mathcal{G},R}(\mathbf{X})]. \tag{30}$$

By Lemma B.2 and Lemma 4.1

$$\mathbb{E}_{\hat{p}}[f] \geq \mathbb{E}_{p^\star}[f] - 2B\varepsilon \geq \Delta_{\min} - 2B\varepsilon = \Delta_{\min}^{\text{eff}}(n). \tag{31}$$

Then, we have

$$\mathbb{E}_{\hat{p}}[f] - s_n \geq \Delta_{\min}^{\text{eff}}(n) - t_n + 2B\varepsilon. \tag{32}$$

Since $2B\varepsilon \geq 0$, from (28), we have

$$\Lambda_n(\mathcal{G}, \mathbf{R}) \leq \frac{p(\mathcal{G}, \mathbf{R})}{p(\mathcal{G}^\star, \mathbf{R}^\star)} \exp\Big\{-n\big(\Delta_{\min}^{\text{eff}}(n) - t_n\big)\Big\}. \tag{33}$$

Let $S_n := \sum_{(\mathcal{G},\mathbf{R}) \in \mathcal{H}} \Lambda_n(\mathcal{G}, \mathbf{R})$. By Assumption (A1), we have $\varepsilon \to 0$ as $n \to \infty$. By Lemma 4.1, $\Delta_{\min} > 0$ is a constant (independent of $n$). Thus, $\Delta_{\min}^{\text{eff}}(n) \to \Delta_{min}$ as $n \to \infty$. $t_n$ also goes to 0 as $n \to \infty$. Thus

$$S_n \leq C \exp\Big\{-n\big(\Delta_{\min}^{\text{eff}}(n) - t_n\big)\Big\} \xrightarrow[n\to\infty]{} 0 \text{ for some constant } C. \tag{34}$$

Since

$$1 = \sum_{(\mathcal{G},\mathbf{R}) \in \mathcal{M}} p(\mathcal{G}, \mathbf{R} \mid \mathcal{D}) = p(\mathcal{G}^\star, \mathbf{R}^\star \mid \mathcal{D}) + \sum_{(\mathcal{G},\mathbf{R} \in \mathcal{H})} p(\mathcal{G}, \mathbf{R} \mid \mathcal{D}) \tag{35}$$

and

$$\sum_{(\mathcal{G},\mathbf{R}), \in \mathcal{H}} p(\mathcal{G}, \mathbf{R} \mid \mathcal{D}) = p(\mathcal{G}^\star, \mathbf{R}^\star \mid \mathcal{D}) \sum_{(\mathcal{G},\mathbf{R}) \in \mathcal{H}} \Lambda_n(\mathcal{G}, \mathbf{R}) = p(\mathcal{G}^\star, \mathbf{R}^\star \mid \mathcal{D})S_n, \tag{36}$$

we have

$$p(\mathcal{G}^\star, \mathbf{R}^\star \mid \mathcal{D}) = \frac{1}{1 + S_n} \geq 1 - S_n, \tag{37}$$

Note that for any two events $W, E$, we have $W = (W \cap E) \cup (W \cap E^c)$. Thus,

$$p(W) = p(W \cap E) + p(W \cap E^c) \leq p(E^c) + p(W \cap E). \tag{38}$$

Fix $\alpha > 0$, let $E_n^c$ be the complement of $E_n$, consider the event $A = \{p(\mathcal{G}^\star, \mathbf{R}^\star \mid \mathcal{D}) \leq 1 - \alpha\}$, Applying (38) gives us

$$p(A) \leq p(E_n^c) + p(A \text{ and } E_n) \tag{39}$$

Given (37) and $p(\mathcal{G}^\star, \mathbf{R}^\star \mid \mathcal{D}) \leq 1 - \alpha$ , we have that

$$S_n \geq \alpha, \tag{40}$$

which means

$$A \cap E_n \subseteq \{S_n \geq \alpha\} \cap E_n. \tag{41}$$

Combining with (39) gives us

$$p(A) \leq p(E_n^c) + p(\{S_n \geq \alpha\} \text{ and } E_n) \tag{42}$$

From (34), we know that for large enough $n$, $S_n < \alpha$, which implies

$$p(\{S_n \geq \alpha\} \text{ and } E_n) = 0 \text{ for large enough } n. \tag{43}$$

Hence,

$$p(A) \leq p(E_n^c) = \eta + \delta/2 \xrightarrow[n\to\infty]{} 0. \tag{44}$$

Therefore,

$$p(\mathcal{G}^\star, \mathbf{R}^\star \mid \mathcal{D}) \xrightarrow[n\to\infty]{p^\star} 1. \tag{45}$$

## B.4. Proof of Theorem 4.4

**Theorem 4.4** (Finite-sample bound with $\varepsilon$-robustness). *For any $\delta \in (0, 1)$, let $M$ be the number of wrong pairs $(\mathcal{G}, \mathbf{R})$ considered and let $\Delta_{\min}^{\text{eff}}(n) := \Delta_{\min} - 2B\varepsilon$, $t_n := B\sqrt{\frac{2\ln(2M/\delta)}{n}}$. With probability at least $1 - \delta - \eta$,*

$$p(\mathcal{G}^\star, \mathbf{R}^\star \mid \mathcal{D}) \geq 1 - M \exp\{-n(\Delta_{\min}^{\text{eff}}(n) - t_n)\}$$
$$\times \max_{(\mathcal{G}, \mathbf{R}) \neq (\mathcal{G}^\star, \mathbf{R}^\star)} \frac{p(\mathcal{G}, \mathbf{R})}{p(\mathcal{G}^\star, \mathbf{R}^\star)}. \tag{7}$$

*Proof.* Let $\mathcal{M}$ be a finite collection of model indices, each of the form $(\mathcal{G}, \mathbf{R})$, where $\mathcal{G}$ is a causal graph and $\mathbf{R}$ is a set of root causes. For each $(\mathcal{G}, \mathbf{R}) \in \mathcal{M}$, let $p_{\mathcal{G}, \mathbf{R}} = p(x \mid \mathcal{G}, \mathbf{R})$ denote the corresponding probability density. To simplify notations, we write $f := f_{\mathcal{G}, \mathbf{R}} = \log(p_{\mathcal{G}^\star, \mathbf{R}^\star} / p_{\mathcal{G}, \mathbf{R}})$, so $|f| \leq B$ by the Assumption (A2).

For any $t > 0$, Hoeffding's inequality yields

$$p\left(\frac{1}{n}\sum_{i=1}^{n} f(\mathbf{X}_i) < \mathbb{E}_{p^\star}[f(\mathbf{X})] - t\right) \leq \exp\left\{-\frac{2nt^2}{(2B)^2}\right\}. \tag{46}$$

By Lemma B.2, on the event $\{d_{\text{TV}}(\hat{p}, p^\star) \leq \varepsilon_n\}$ we have

$$\mathbb{E}_{\hat{p}}[f] \geq \mathbb{E}_{p^\star}[f] - 2B\varepsilon \iff \mathbb{E}_{p^\star}[f] \leq \mathbb{E}_{\hat{p}}[f] + 2B\varepsilon. \tag{47}$$

Combining (46) and (19), on $\{d_{\text{TV}}(\hat{p}, p^\star) \leq \varepsilon\}$ we obtain

$$p\left(\frac{1}{n}\sum_{i=1}^{n} f(\mathbf{X}_i) < \mathbb{E}_{\hat{p}}[f] + 2B\varepsilon - t \;\Big|\; d_{\text{TV}}(\hat{p}, p^\star) \leq \varepsilon\right) \leq \exp\left\{-\frac{2nt^2}{(2B)^2}\right\}. \tag{48}$$

Equivalently,

$$p\left(\frac{1}{n}\sum_{i=1}^{n} f(\mathbf{X}_i) \geq \mathbb{E}_{\hat{p}}[f] - (t - 2B\varepsilon_n) \;\Big|\; d_{\text{TV}}(\hat{p}, p^\star) \leq \varepsilon\right) \geq 1 - \exp\left\{-\frac{2nt^2}{(2B)^2}\right\}. \tag{49}$$

Let

$$t_n := B\sqrt{\frac{2\ln(2M/\delta)}{n}}, \text{ where } M = |\mathcal{H}| \qquad \text{and} \qquad s_n := t_n - 2B\varepsilon.$$

Since $\varepsilon \to 0$, we have $s_n > 0$ for all large $n$. Plugging $t = t_n$ into (21) gives, for each fixed $(\mathcal{G}, \mathbf{R}) \in \mathcal{H} := \mathcal{M}\setminus\{(\mathcal{G}^\star, \mathbf{R}^\star)\}$, for any $\delta \in (0,1)$

$$p\left(\frac{1}{n}\sum_{i=1}^{n} f_{\mathcal{G},\mathbf{R}}(X_i) \geq \mathbb{E}_{\hat{p}}[f] - s_n \ \bigg| \ d_{\mathrm{TV}}(\hat{p}, p^\star) \leq \varepsilon\right) \geq 1 - \frac{\delta}{2M}. \tag{50}$$

Applying a union bound over $(\mathcal{G}, \mathbf{R}) \in \mathcal{H}$,

$$p\left(\bigcap_{(\mathcal{G},\mathbf{R})\in\mathcal{H}}\left\{\frac{1}{n}\sum_{i=1}^{n} f_{\mathcal{G},\mathbf{R}}(X_i) \geq \mathbb{E}_{\hat{p}}[f] - s_n\right\} \ \bigg| \ d_{\mathrm{TV}}(\hat{p}, p^\star) \leq \varepsilon\right) \geq 1 - \frac{\delta}{2}.$$

Finally, combine with Assumption (A1), $p(d_{\mathrm{TV}}(\hat{p}, p^\star) \leq \varepsilon) \geq 1 - \eta$, to conclude

$$p(E_n) \geq 1 - \eta - \frac{\delta}{2} > 1 - \eta - \delta, \tag{51}$$

where

$$E_n = \left\{d_{\mathrm{TV}}(\hat{p}, p^\star) \leq \varepsilon\right\} \cap \left(\bigcap_{(\mathcal{G},\mathbf{R})\in\mathcal{H}}\left\{\frac{1}{n}\sum_{i=1}^{n} f_{\mathcal{G},\mathbf{R}}(X_i) \geq \mathbb{E}_{\hat{p}}[f_{\mathcal{G},\mathbf{R}}] - s_n\right\}\right).$$

Next, we define

$$\Delta_{\min}^{\mathrm{eff}}(n) := \Delta_{\min} - 2B\varepsilon, \tag{52}$$

where

$$\Delta_{\min} := \inf_{(\mathcal{G},\mathbf{R})\neq(\mathcal{G}^\star,\mathbf{R}^\star)} \mathbb{E}_{p^\star}[f_{\mathcal{G},R}(\mathbf{X})]. \tag{53}$$

By Lemma B.2 and Lemma 4.1

$$\mathbb{E}_{\hat{p}}[f] \geq \mathbb{E}_{p^\star}[f] - 2B\varepsilon \geq \Delta_{\min} - 2B\varepsilon = \Delta_{\min}^{\mathrm{eff}}(n). \tag{54}$$

From (54), we can derive

$$\sum_{i=1}^{n} f_{\mathcal{G},\mathbf{R}}(\mathbf{X}_i) \geq n(\Delta_{\min}^{\mathrm{eff}}(n) - t_n) \tag{55}$$

For each wrong pair $(\mathcal{G}, \mathbf{R}) \in \mathcal{H}$, define

$$\Lambda_n(\mathcal{G}, \mathbf{R}) := \frac{p(\mathcal{G}, \mathbf{R} \mid \mathcal{D})}{p(\mathcal{G}^\star, \mathbf{R}^\star \mid \mathcal{D})} = \frac{p(\mathcal{G}, \mathbf{R})}{p(\mathcal{G}^\star, \mathbf{R}^\star)}\prod_{i=1}^{n}\frac{p(\mathbf{X}_i \mid \mathcal{G}, \mathbf{R})}{p(\mathbf{X}_i \mid \mathcal{G}^\star, \mathbf{R}^\star)}. \tag{56}$$

Then,

$$\Lambda_n(\mathcal{G}, \mathbf{R}) = \frac{p(\mathcal{G}, \mathbf{R})}{p(\mathcal{G}^\star, \mathbf{R}^\star)}\exp\left\{-\sum_{i=1}^{n} f_{\mathcal{G},\mathbf{R}}(\mathbf{X}_i)\right\}. \tag{57}$$

With (55), we have

$$\Lambda_n(\mathcal{G}, \mathbf{R}) \leq \frac{p(\mathcal{G}, \mathbf{R})}{p(\mathcal{G}^\star, \mathbf{R}^\star)}\exp\left\{-n(\Delta_{\min}^{\mathrm{eff}}(n) - t_n)\right\}. \tag{58}$$

Summing over $\mathcal{H}$,

$$S_n := \sum_{(\mathcal{G},\mathbf{R})\in\mathcal{H}} \Lambda_n(\mathcal{G},\mathbf{R}) \leq \exp\left\{-n(\Delta_{\min}^{\text{eff}}(n) - t_n)\right\} \sum_{(\mathcal{G},\mathbf{R})\in\mathcal{H}} \frac{p(\mathcal{G},\mathbf{R})}{p(\mathcal{G}^\star,\mathbf{R}^\star)} \tag{59}$$

$$\leq M \exp\left\{-n(\Delta_{\min}^{\text{eff}}(n) - t_n)\right\} \max_{(\mathcal{G},\mathbf{R})\in\mathcal{H}} \frac{p(\mathcal{G},\mathbf{R})}{p(\mathcal{G}^\star,\mathbf{R}^\star)}. \tag{60}$$

Since

$$1 = \sum_{(\mathcal{G},\mathbf{R})\in\mathcal{M}} p(\mathcal{G},\mathbf{R} \mid \mathcal{D}) = p(\mathcal{G}^\star,\mathbf{R}^\star \mid \mathcal{D}) + \sum_{(\mathcal{G},\mathbf{R}\in\mathcal{H})} p(\mathcal{G},\mathbf{R} \mid \mathcal{D}) \tag{61}$$

and

$$\sum_{(\mathcal{G},\mathbf{R}),\in\mathcal{H}} p(\mathcal{G},\mathbf{R} \mid \mathcal{D}) = p(\mathcal{G}^\star,\mathbf{R}^\star \mid \mathcal{D}) \sum_{(\mathcal{G},\mathbf{R})\in\mathcal{H}} \Lambda_n(\mathcal{G},\mathbf{R}) = p(\mathcal{G}^\star,\mathbf{R}^\star \mid \mathcal{D})S_n, \tag{62}$$

we have

$$p(\mathcal{G}^\star,\mathbf{R}^\star \mid \mathcal{D}) = \frac{1}{1 + S_n} \geq 1 - S_n, \tag{63}$$

Put (60) into (63) to obtain

$$p(\mathcal{G}^\star,\mathbf{R}^\star \mid \mathcal{D}) \geq 1 - M \exp\left\{-n(\Delta_{\min}^{\text{eff}}(n) - t_n)\right\} \max_{(\mathcal{G},\mathbf{R})\in\mathcal{H}} \frac{p(\mathcal{G},\mathbf{R})}{p(\mathcal{G}^\star,\mathbf{R}^\star)}. \tag{64}$$

Since every outcome in $E_n$ satisfies the posterior in (64), $E_n \subseteq A_n$, where $A_n = \{p(\mathcal{G}^\star,\mathbf{R}^\star \mid \mathcal{D}) \geq 1 - M \exp\{-n(\Delta_{\min}^{\text{eff}}(n) - t_n)\} \max_{(\mathcal{G},\mathbf{R})\in\mathcal{H}} \frac{p(\mathcal{G},\mathbf{R})}{p(\mathcal{G}^\star,\mathbf{R}^\star)}\}$. Hence, $p(A_n) \geq p(E_n)$. By (51),

$$p(A_n) \geq 1 - \eta - \delta. \tag{65}$$

$\square$

### B.5. Proof of Lemma 4.5

*Proof.* First, notice that $(\mathcal{G},\mathbf{R})$ can be partitioned according to the interventional distributions they map to. Lemma 4.1 shows that each realization of $\mathbf{R}$ corresponds to a different interventional distribution. If we consider $\mathbf{R} = \mathbf{r}$, we want to show that each configuration of $\mathbf{r}$ refers to a unique distribution of $P(\mathcal{D}|\mathcal{G},\mathbf{r})$. We would like to introduce a helpful lemma below.

**Lemma B.3.** *(Elahi et al., 2024) Assume that the faithfulness assumption holds and $\mathcal{D}^*$ is the true DAG. For any DAG $\mathcal{D}_1 \neq \mathcal{D}^*$, if $P_{\mathbf{s}}^{\mathcal{D}_1} = P_{\mathbf{s}}^{\mathcal{D}^*}$ for some $\mathbf{S} \subseteq \mathbf{V}$, they must share the same cutting edge orientation $Q(\mathbf{S})$.*

Consider each cutting edge configuration $Q(\mathbf{r})$ at $\mathbf{r}$. For any $\mathcal{G} \in \mathcal{C}_{Q(\mathbf{r})}(\mathcal{G}^*)$, Lemma B.3 shows that they correspond to the same distribution. Hence, we can partition $(\mathcal{G},\mathbf{R})$ based on the choice of $\mathbf{R}$ and the cutting edge configuration of $\mathbf{R}$, which are unique $\mathcal{I}$-CPDAGs. With a uniform prior, ranking the posteriors of $(\mathcal{G},\mathbf{R})$ is the same as ranking them for the $\mathcal{I}$-CPDAGs. $\square$

## C. Description of the Algorithms used for Experiments

We provide a brief description of the algorithms used in various experiments below.

- **BARO** (Pham et al., 2024) It takes the multivariate time series metrics and the estimated anomaly time. For each metric, it learns the observational distribution by analyzing the data points before the detected anomaly time by estimating the central tendency and spread using the median and interquartile range (IQR). For each data point after the anomaly time, it computes the absolute difference between each data point and the median rescaled by IQR to determine the top root causes.

- **RCG** (Ikram et al., 2025) Given a $k$-essential graph (Kocaoglu, 2023) or a CPDAG, it applies marginal independence tests on each variable with $F$, then it computes the conditional mutual information between each variable given a set of possible parents in the graph and ranks the root cause variables in descending order.

- **RCD** (Ikram et al., 2022) It first randomly partitions all the metrics. Then, it uses conditional independence tests to repeatedly test the independence between the binary indicator variable $F$ and each metric, conditioning on the subsets of the variables within each partition. The metrics that are still found dependent after the series of CI tests will be put into the same partition. This process repeats until it finds the top $k$ root causes ranked by the p-values associated with the CI tests in ascending order.

- **SCORE ORDERING (SO)** (Orchard et al., 2025) It applies an outlier scoring function $\pi$ such as z-score for a given anomalous sample $x_j$ and then computes the fraction of samples from normal data whose scores are greater than or equal to $\pi(x_j)$ and divides that by the total number of samples. Then, it takes a negative logarithm of this ratio. The smaller the ratio, the higher the score becomes. The node with the highest score will be output as the root cause. It does not require any causal graph.

- **SMOOTH TRAVERSAL (ST)** (Orchard et al., 2025) assumes a causal DAG and the knowledge of the trigger point are available. It traverses from the trigger point variable to its ancestors and determines whether a variable is the root cause by checking the largest difference between the variable and its parents in marginal anomaly score.

- **In-Distribution Interventions (IDI)** (Nagalapatti et al., 2025) learns anomaly scores and an SCM on normal data over a known DAG. It selects candidates that are anomalous while their parents are normal, then evaluates each node by *in-distribution interventions* that resample it from $P(X_i \mid \text{Pa}(X_i))$ for each node $X_i$ and propagate forward to measure the target anomaly reduction. Finally, it uses Shapley values to rank nodes by attributed impact.

- **CausalRCA** (Xin et al., 2023) It first constructs a weighted causal graph between metrics based on anomalous samples using a gradient-based structure learning method named DAG-GNN (Yu et al., 2019), which can capture both linear and nonlinear dependencies. Once the causal graph is learned, PageRank is applied in reverse on the graph to rank metrics by their likelihood of being the root cause of an observed anomaly.

- **CIRCA** (Li et al., 2022) uses PC algorithm (Spirtes et al., 2001) to obtain the adjacency of the observed metrics. It then uses regression-based hypothesis testing by training a regression model for each node $V_i$ on normal operation data to predict $V_i$ from the parents of $V_i$ based on the estimated $P_N$. This is motivated by the observation that $P_N(R|Pa(R)) \neq P_A(R|Pa(R))$ for any root cause variable $R$, where $P_A$ is the post-interventional distribution.

- **Cholesky** (Li et al., 2025) assumes a linear structural causal model but does not require knowledge of the causal graph. It identifies the root cause by applying a Cholesky decomposition to the covariance matrix of the observed variables and exploiting a permutation-invariance property.

- **SimpleRCA** (Fang et al., 2025) takes the 95th percentile of the distribution from the normal dataset and takes the difference between that and the maximum of the distribution of the anomalous dataset to determine the true root cause.

- **ShapleyIQ** (Li et al., 2024) first builds a causal graph from traces, metrics, and logs, then uses physical counterfactual models such as latency propagation, queueing behavior, QPS effects to estimate how the global metric would change if a subset of factors became abnormal. It then assigns influence scores using a modified Shapley value. For our experiments, since we do not use any trace data, we modify the algorithm to adapt to our setting. We fit one model per node based on the given service call graph using ridge regression. We then compute the residual abnormal shift per node as follows: we take the median $m_N$ from the normal data and the median $m_A$ from the abnormal data. We then compute the total shift by $m_A - m_N$ and the inherited shift by $f_j(m_A) - f_j(m_N)$ and compute the residual shift by subtracting the inherited shift from the total shift. It then takes the set of ancestors of the target node and ranks the nodes inside the set. The way to rank them is to compute the marginal contribution of the selected node to the normal value given by $f_j(Pa_j)$ for each observed variable $X_j$.

- **MicroDig** (Tao et al., 2024) first identifies the calls associated with an SLO violation using port-level call data and anomaly detection, then aggregates them into a service-level association graph. It transforms this graph into a heterogeneous propagation graph where both services and calls are nodes: service nodes point to their related call nodes, and downstream call nodes point to upstream call nodes to model anomaly propagation. Finally, it runs a Heterogeneity-Oriented Random Walk (HORW) that uses correlation between call anomaly-rate time series and service

anomaly scores to rank candidate root-cause services. We modify the algorithm to adapt to our experimental setup so that it can operate on tabular metric data instead of the paper's aggregated call-level data. We omit key upstream steps such as association-call extraction, issue-time-specific graph construction, and anomaly-based node filtering due to the lack of trace data. Instead, it computes per-node anomaly scores using a z-score shift heuristic, which are used as weights rather than as a mechanism to prune non-anomalous nodes. Although it preserves the idea of a heterogeneous graph with service and call nodes and applies a random-walk-style ranking, it relies on a standard personalized PageRank-like propagation instead of the paper's HORW formulation with its $\beta$ parameter, given that we cannot detect correlation between calls based on metric data.

- **BRCD-C** is a variant of **BRCD** that takes the proxy causal graph constructed based on a service map of the metric as the CPDAG input.

- **BRCD-M** is a variant of **BRCD** that incorporates the background knowledge from the service map by only considering the CPDAGs where it has all the ancestral relations consistent with the background knowledge in the causal discovery algorithm BOSS (Andrews et al., 2023).

- **BRCD-B10** is a variant of **BRCD** where it first bootstraps the observational data ten different times to estimate the posterior of CPDAGs and incorporates that posterior in **BRCD**.

- **BRCD-B100** is a variant of **BRCD** where it first bootstraps the observational data a hundred different times to estimate the posterior of CPDAGs and incorporates that posterior in **BRCD**.

## D. Implementation Details for Experiments

**Synthetic experiment**   In the synthetic experiment, **BRCD** and **RCG** use a ground truth CPDAG. **RCG** computes the conditional mutual information using frequency count on the discrete data. **RCD** uses chi-sq tests with a starting $\alpha = 0.01$. For **BRCD**, we train a quantile discretization model with normal data with $k$-binning where $k = 5$. We then assign the abnormal samples to the states based on the trained model. We conduct the graph sampling scheme in parallel.

**Real-world data experiment**   We use the default setting of the causal discovery algorithm named BOSS (Andrews et al., 2023) from `causal-learn` (Zheng et al., 2024) in Python to obtain the CPDAG for **BRCD** and its variants. We handle the case whenever BOSS throws any error due to a singular matrix for computing BIC score by removing any metrics that have zero variance. We can confirm that such metrics are never the true root causes. We also use cache to avoid recomputing the likelihood for each configuration. We have used a linear Gaussian likelihood estimator for **BRCD** and its variants as discussed in Appendix F. For **CIRCA**, **CausalRCA**, **BARO**, **RCD**, we use the same default setup and codes from RCAEval GitHub repository provided by Pham et al. (2025). For **SO** and **ST**, we use the code provided by Orchard et al. (2025). We use **IDI** based on the Github repository provided by (Nagalapatti et al., 2025) with the same default parameters. For **RCG**, **ST**, and **IDI**, they are given the proxy causal graph by reversing the given service map. The ancestral relations of the metrics follow their respective service depicted by the service map. More specifically, each metric under service $A$ will have a directed edge to another metric under service $B$ whenever there is a directed edge from service $A$ and service $B$. We also create a variant of **BRCD-C** where it takes the proxy causal graph as the CPDAG input. For OB experiments, **ST** and **IDI** are given the frontend metric with respect to the fault type as the trigger point. For Sockshop experiment, the trigger point variable is `front-end_container_cpu` whenever the fault type is not memory leak as there is no known trigger point provided by the dataset. Otherwise, it will be `front-end_container_mem`. The service map of OB experiment follows Figure 4. We follow the service map of the Sockshop application provided by Orchard et al. (2025) as shown in Figure 5. For both OB and Sockshop experiments, we follow the same data preprocessing steps as provided by (Pham et al., 2025). For the Petshop dataset, we remove any variable that only contains missing values. We also impute the remaining variables by the median within the normal data for the missing value in the normal and the median of abnormal data for any missing value in abnormal data. We did not provide the result of **CIRCA** as it takes an extensive amount of time to run the PC algorithm as part of the algorithm. We also cannot obtain results for **CausalRCA** on Petshop, as it throws a numerical error during the model training.

All the RCA methods are run on a cluster with an NVIDIA GeForce RTX 3090 graphics card, which has a total of 24 GB of memory. The cluster has 128 CPUs with 126 GB of RAM.

### D.1. Service Map for Online Boutique (Google Cloud Platform)

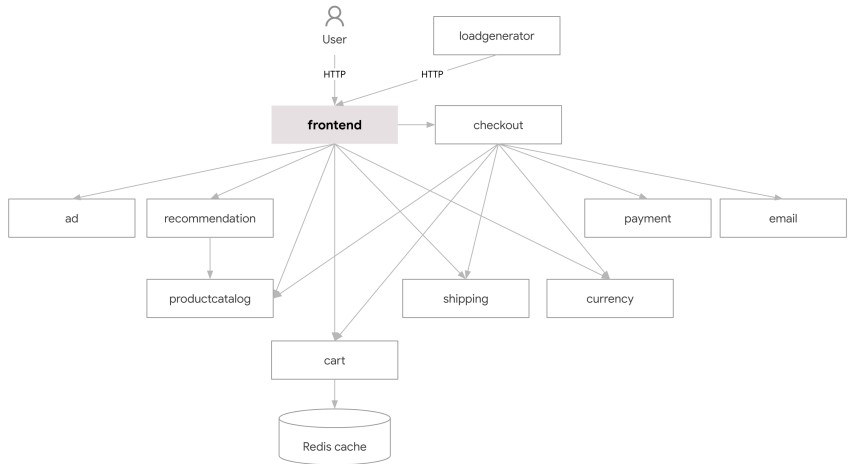

*Figure 4.* Online Boutique Service Map

### D.2. Service Map for Sockshop provided by Orchard et al. (2025)

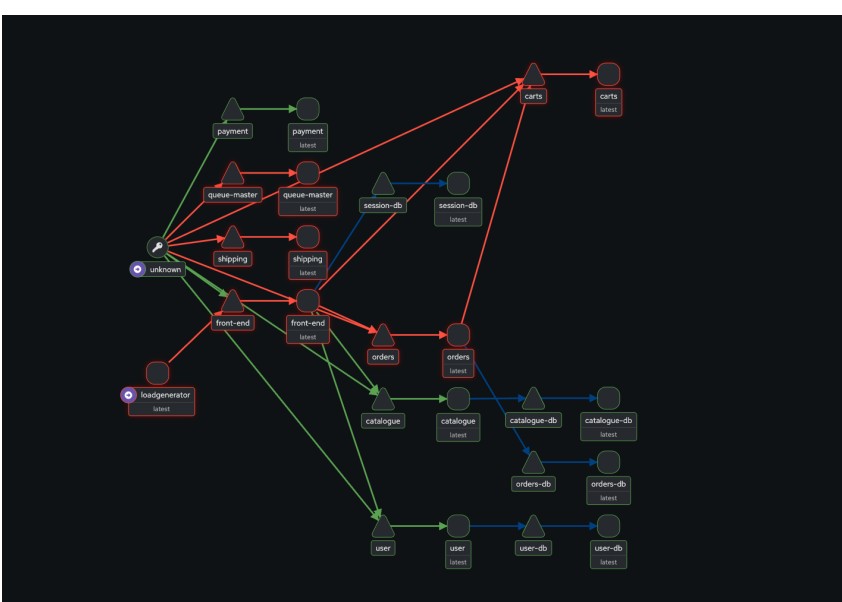

*Figure 5.* Service Map of Sockshop application

## E. Time Complexity of BRCD

In this section, we give an analysis on the time complexity of the proposed algorithm. For simplicity, we will analyze under the single root cause case for discrete data. Given a CPDAG $\mathcal{C}(\mathcal{G}^\star) = (\mathbf{V}, \mathbf{E})$, let $n = |\mathbf{V}|$ be the number of nodes, $m$ be the total number of edges, $u$ be the number of undirected edges, $d_u(V)$ be the number of undirected neighbors of the node $V$ and $d_{\max} = \max_V d_u(V)$. Let $N$ be the total number of samples and $r_i$ be the number of discrete states of $X_i$ and $r_{max} = \max_i r_i$. We also let $q_i = \prod_{j \in Pa(i)} r_j$ be the number of parent configurations of $X_i$ in a sampled DAG.

For each node $V$, **BRCD** enumerates all directions for the incident undirected edges i.e. $2^{d_u(V)}$ local configurations. The total configurations considered will be $Q = \sum_{V_i} 2^{d_u(V_i)} \leq n2^{d_{\max}}$. Next, we discuss per-configuration costs. For each configuration, we apply a set of graphical orientation rules known as Meek rules, which takes $\mathcal{O}(nd_{\max}^2)$

(Meek, 1995). We then check for acyclicity with depth first search, which takes $\mathcal{O}(n+m)$. We also check if there exists any new unshielded colliders, which will be bounded by $\mathcal{O}(nd_{\max}^2)$. Once we have a configuration ready as a $\mathcal{I}$-CPDAG, we then apply the polynomial-time DAG sampling algorithm to an augmented graph, which takes $\mathcal{O}(n^4)$ (Wienöbst et al., 2023). Then, for an augmented graph, computing the Dirichlet posterior-predictive, we only need to count the number of states for each parent configuration per node, in the worse case, it takes $\mathcal{O}(N)$ per node with hash maps, yielding $\mathcal{O}(nN)$ across all nodes. The likelihood evaluation will then be $\mathcal{O}(\sum_i^n r_i q_i)$. Thus, the total complexity per configuration will be $\mathcal{O}(nN + nd_{\max}^2 + m + n^4 + \sum_i^n r_i q_i)$. Therefore, the overall time complexity will be $\mathcal{O}\left(\sum_{V_i} 2^{d_u(V_i)}(nN + nd_{\max}^2 + m + n^4 + \sum_i^n r_i q_i)\right)$. In the worst case, **BRCD** is exponential in the local undirected degree due to enumerating $2^{d_u(V)}$ configurations per node and polynomial in $(n, m)$ and the data factor $N$ per configuration, with an extra exponential dependence on max in-degree inside the Dirichlet evaluation. However, we note that many of these computations can be performed during normal operation. The only computational expense during the failure period is the likelihood evaluation that involves $F$ as a parent.

## F. Practical guidance on selecting estimators for the likelihood estimation during posterior update

In this section, we provide some suggestions on what estimators to use for estimating the likelihood during the posterior update. Note that our posterior consistency and finite sample bound proofs do not depend on which estimator to use.

For discrete data, we adopt the Dirichlet posterior-predictive (prequential) likelihood as our default scoring rule for our discrete conditional probability table after discretization to avoid the optimism of plug-in likelihoods that both fit and evaluate on the same data. Concretely, for each family $p(X \mid Pa(X))$, we place a symmetric Dirichlet prior with equivalent sample size $\alpha^\star$ and score each observation with the posterior predictive as follows.

**Dirichlet prequential scoring** Fix a discrete node $X$ with $K$ states and parents $Pa(X)$. For each parent configuration $u \in \mathcal{S}(Pa(X))$, where $\mathcal{S}(Pa(X))$ is the set of states for parents of $X$, let the row parameters be $\theta_{\cdot|u} \in \Delta^{K-1}$ with multinomial likelihood $p(x \mid u, \theta) = \prod_{x=1}^{K} \theta_{x|u}^{\mathbf{1}\{X=x\}}$. We place a *symmetric Dirichlet* prior on each $u$:

$$\theta_{\cdot|u} \sim Dir\Big(\underbrace{\tfrac{\alpha^\star/q}{K}, \ldots, \tfrac{\alpha^\star/q}{K}}_{K \text{ entries}}\Big) \tag{66}$$

with $q = \prod_{Z \in Pa(X)} K_Z$ the number of parent configurations and $K_Z$ the cardinality of parent $Z$. Given data $\mathcal{D}$, let $N_{x|u}$ be the count of $X = x$ when $Pa(X) = u$ and $N_{\cdot|u} = \sum_x N_{x|u}$. The *posterior* for each row is

$$\theta_{\cdot|u} \mid \mathcal{D} \sim Dir\Big(\tfrac{\alpha^\star/q}{K} + N_{1|u}, \ldots, \tfrac{\alpha^\star/q}{K} + N_{K|u}\Big),$$

and the prequential probability for a new outcome $x$ under $u$ is

$$p(x \mid u, \mathcal{D}) = \frac{N_{x|u} + \alpha_{\text{row}}/K}{N_{\cdot|u} + \alpha_{\text{row}}}. \tag{67}$$

When scoring the dataset within each $u$, we multiply these prequential terms. The product over all observations equals the *Dirichlet–multinomial marginal likelihood*, also known as Dawid's prequential identity (Dawid, 1984):

$$\prod_u \frac{\Gamma(\alpha^\star/q)}{\Gamma((\alpha^\star/q) + N_{\cdot|u})} \prod_{x=1}^{K} \frac{\Gamma((\alpha^\star/q)/K + N_{x|u})}{\Gamma((\alpha^\star/q)/K)} \tag{68}$$

In short, for each $u$, we shrink sparse counts by $(\alpha^\star/q)/K$ and evaluate each observation by the fraction above, aggregating across rows in a discrete conditional probability table to yield the standard BDeu integrated score with an automatic Occam penalty and strict positivity. This choice guarantees positivity and stabilizes thin-count regimes induced by discretization and many parents, and remains locally decomposable so we can compare graphs family-wise and include the environment indicator $F$ as a parent to test invariance vs. shift via Bayes factors. We keep $\alpha^*$ fixed as a hyperparameter and fit discretization bins on normal data ($F = 0$). Empirically and theoretically, this integrated score preserves our identifiability and posterior-consistency guarantees while improving finite-sample robustness.

For continuous data, which is more common in microservices data, instead of scoring with a Dirichlet posterior–predictive. One can explore the dataset to see if any parametric assumption can be made in order to gain computational efficiency while maintaining high accuracy. For example, in the real-world experiments, we model each continuous family $p(X \mid Pa(X))$ as a linear Gaussian conditional on a possibly transformed scale.

**Optional Transform**  For a continuous child $X$, we optionally apply a monotone transform $z = T(X)$ such as $\log$, $\log(1+x)$. All parents may optionally receive the same transform to improve linearity. Because the node is transformed, the conditional density on the original scale includes the Jacobian:

$$p(x \mid \cdot) \;=\; p_Z\big(T(x) \mid \cdot\big)\left|\tfrac{d\,T(x)}{dx}\right|. \tag{69}$$

**Linear Gaussian Conditional Probability Density Estimation**  Let $U$ denote the continuous parents of $X$ other than $F$. In each regime $F = f \in \{0,1\}$, we posit

$$z \;=\; \beta_{f,0} + \beta_f^\top U \;+\; \varepsilon_f, \qquad \varepsilon_f \sim \mathcal{N}(0, \sigma_f^2). \tag{70}$$

We place a conjugate Normal Inverse Gamma prior on $(\beta_f, \sigma_f^2)$ and use the posterior predictive based on Student–t distribution for scoring. Let $X_f \in \mathbb{R}^{n_f \times p}$ be the design matrix with a leading 1 (for the intercept), $y_f \in \mathbb{R}^{n_f}$ the transformed responses, and hyperparameters $(m_0, \Lambda_0, \alpha_0, \beta_0)$. The posterior is

$$\beta_f^{(\text{post})} = \beta_0 + \tfrac{1}{2}\big(y_f^\top y_f + m_0^\top \Lambda_0 m_0 - m_f^\top \Lambda_f m_f\big), \tag{71}$$

where

$$\Lambda_f = \Lambda_0 + X_f^\top X_f, \quad m_f = \Lambda_f^{-1}(\Lambda_0 m_0 + X_f^\top y_f), \quad \alpha_f = \alpha_0 + \tfrac{n_f}{2}. \tag{72}$$

For a new row with covariate $x_\star$ (including the intercept),

$$z_\star \mid \mathcal{D}_f \sim \text{Student-}t\Big(\nu_f = 2\alpha_f,\ \mu_f = x_\star^\top m_f,\ s_f^2 = \tfrac{\beta_f^{(\text{post})}}{\alpha_f}\big(1 + x_\star^\top \Lambda_f^{-1} x_\star\big)\Big). \tag{73}$$

The per-row predictive on the untransformed scale is then

$$p(x_\star \mid U_\star, F = f, \mathcal{D}) \;=\; t_{\nu_f}\big(T(x_\star); \mu_f, s_f^2\big)\left|\tfrac{d\,T(x_\star)}{dx}\right|. \tag{74}$$

Multiplying these over rows (in any order) yields the integrated marginal likelihood for the family.

This continuous prequential scoring rule remains strictly positive, locally decomposable, and compatible with Bayesian updating over DAG samples used by BRCD from each possible $\mathcal{I}$-CPDAG. It also preserves our identifiability and posterior-consistency guarantees under mild regularity.

## G. Additional Experiments

For both OB and Sockshop, we use the same evaluation metric as in Pham et al. (2025), that is, $Avg@k = \frac{1}{k}\sum_{j=1}^k AC@j$, where $AC@j$ denotes the probability the top $k$ results of the given method include the true root causes. More specifically, Pham et al. (2025) defines $AC@j = \frac{1}{A}\sum_{a \in A}\frac{\sum_{i \le k} R^a[i] \in V_{rc}^a}{\min(k, |V_{rc}^a|)}$, where $R^a[i]$ is the ith ranking result for the failure case $a$ by an RCA algorithm and $V_{rc}^a$ is the true root cause set of case $a$. We also evaluate the algorithms via Mean Reciprocal Rank (MRR), which measures how highly the true root cause appears in the ranked list of candidate root causes produced by an algorithm. We show the result in terms of MRR in Table 6.

### G.1. Online Boutique

**Online Boutique (OB)** is Google's 12-service e-commerce demo where users browse, add to cart, and purchase items. Microservices communicate primarily over gRPC. In the RCAEval benchmark (Pham et al., 2025), OB is instrumented with metrics and includes injected resource and network faults organized into failure cases. We will examine each fault

| | | CIRCA | RCG | BARO | RCD | IDI | ST | SO | CausalRCA | Cholesky | BRCD-C | BRCD-B10 | BRCD | SimpleRCA | ShapleyIQ | MicroDig |
|---|---|---|---|---|---|---|---|---|---|---|---|---|---|---|---|---|
| Avg@1 | CPU | 0.24 | 0.72 | **0.92** | 0.84 | 0.64 | 0.08 | 0.00 | 0.20 | 0.12 | 0.88 | **0.92** | **0.92** | 0.20 | **0.92** | 0.00 |
| | MEM | 0.20 | 0.64 | 0.96 | 0.64 | 0.12 | 0.08 | 0.00 | 0.20 | 0.00 | **1.00** | 0.92 | 0.96 | 0.76 | 0.80 | 0.00 |
| | DISK | 0.72 | **0.76** | 0.72 | 0.72 | 0.36 | 0.12 | 0.16 | 0.20 | 0.08 | **0.76** | 0.60 | 0.64 | 0.76 | 0.48 | 0.00 |
| | DELAY | **0.72** | 0.40 | 0.64 | 0.24 | 0.64 | 0.16 | 0.16 | 0.00 | 0.00 | 0.52 | 0.20 | 0.28 | **1.00** | 0.28 | 0.00 |
| | LOSS | 0.32 | 0.24 | **0.44** | 0.28 | **0.44** | 0.12 | 0.12 | 0.00 | 0.08 | **0.44** | 0.32 | 0.28 | **0.52** | 0.28 | 0.00 |
| | Average | 0.44 | 0.55 | **0.74** | 0.54 | 0.44 | 0.11 | 0.09 | 0.12 | 0.06 | 0.72 | 0.59 | 0.62 | 0.65 | 0.55 | 0.00 |
| Avg@3 | CPU | 0.44 | 0.77 | **0.97** | 0.84 | 0.72 | 0.21 | 0.13 | 0.40 | 0.27 | 0.93 | 0.95 | **0.97** | 0.41 | **0.97** | 0.61 |
| | MEM | 0.40 | 0.75 | 0.99 | 0.67 | 0.12 | 0.27 | 0.08 | 0.40 | 0.12 | **1.00** | 0.97 | 0.99 | 0.92 | 0.92 | 0.61 |
| | DISK | 0.88 | 0.79 | 0.89 | 0.77 | 0.37 | 0.25 | 0.21 | 0.40 | 0.20 | 0.87 | 0.80 | 0.81 | **0.92** | 0.81 | 0.45 |
| | DELAY | 0.87 | 0.64 | 0.87 | 0.27 | 0.69 | 0.25 | 0.15 | 0.20 | 0.13 | 0.63 | 0.32 | 0.41 | **1.00** | 0.47 | 0.40 |
| | LOSS | 0.45 | 0.47 | 0.52 | 0.27 | 0.52 | 0.27 | 0.19 | 0.20 | 0.16 | 0.59 | 0.52 | 0.49 | **0.64** | 0.48 | 0.35 |
| | Average | 0.61 | 0.68 | **0.84** | 0.59 | 0.48 | 0.25 | 0.15 | 0.32 | 0.18 | 0.80 | 0.71 | 0.73 | 0.78 | 0.73 | 0.48 |
| Avg@5 | CPU | 0.60 | 0.78 | **0.98** | 0.86 | 0.74 | 0.34 | 0.22 | 0.56 | 0.34 | 0.94 | 0.95 | **0.98** | 0.57 | **0.98** | 0.75 |
| | MEM | 0.56 | 0.77 | 0.99 | 0.72 | 0.12 | 0.40 | 0.17 | 0.56 | 0.21 | **1.00** | 0.98 | 0.99 | 0.95 | 0.95 | 0.77 |
| | DISK | 0.56 | 0.79 | 0.94 | 0.79 | 0.40 | 0.38 | 0.31 | 0.56 | 0.34 | 0.91 | 0.86 | 0.87 | **0.95** | 0.89 | 0.66 |
| | DELAY | 0.92 | 0.70 | 0.92 | 0.30 | 0.70 | 0.34 | 0.23 | 0.36 | 0.22 | 0.73 | 0.46 | 0.53 | **1.00** | 0.59 | 0.64 |
| | LOSS | 0.58 | 0.57 | 0.65 | 0.54 | 0.55 | 0.39 | 0.30 | 0.36 | 0.29 | 0.69 | 0.60 | 0.58 | **0.74** | 0.59 | 0.58 |
| | Average | 0.64 | 0.72 | **0.90** | 0.64 | 0.50 | 0.37 | 0.25 | 0.48 | 0.28 | 0.85 | 0.77 | 0.79 | 0.84 | 0.80 | 0.68 |

*Table 4.* Avg@k performance on RCAEval Online Boutique RE1 dataset (Pham et al., 2025).

type in this application: CPU hog (CPU), memory leak (MEM), disk stress (DISK), delay variations (DELAY), and failed requests (LOSS). The faults are injected into five services separately: ads, carts, checkout, currency, and product catalog. For each service, five datasets are collected for five different independent injections. For CPU and MEM, each dataset has roughly 50 continuous metric variables, 2100 observational samples, and 2100 failure samples. For DISK, DELAY, and LOSS, there are roughly 360 observational samples, and 360 failure samples in each dataset. We provide the results in Table 4. We see that **BRCD** performs on par with **BARO** in the CPU and MEM faulty types, but **BRCD** tends to underperform in DISK, DELAY, and LOSS. One reason is that DISK, DELAY, and LOSS faulty types have fewer observational samples. This poses challenges to observational causal discovery, which can negatively affect **BRCD**. We observe that **BRCD-C** outperforms **BRCD**, as it leverages a proxy causal graph despite having only a limited number of observed samples. This suggests **BRCD**'s primary bottleneck in real traces is CPDAG quality under limited observational samples rather than the posterior update mechanism itself. **BRCD-B10** also does not improve **BRCD** in this case.

### G.2. Sockshop

The **Sockshop** application, originally developed by Weaveworks, is an e-commerce application for selling socks, implemented as 15 microservices that interact through HTTP requests (OCP Power Demos). Mainly, there are 5 different services injected with faults: carts, user, payment, orders, and catalog. Each of these fault types has 5 different datasets. There are roughly 360 observational samples and 360 anomalous samples. Each dataset contains 59 continuous metric variables. For both OB and Sockshop, we retrieve the RE1 datasets from RCAEval benchmark and follow the same data preprocessing steps in Pham et al. (2025). The results are shown in Table 5. Overall, the performances of the RCA algorithms are similar to those in the OB experiment. We observe that **IDI** significantly suffers in the experiment. We attribute that to a potential mispecification of the trigger point and the complex causal structure of the metrics, even when the service map is provided. We can also see that the decrease in performance also happens to **ST**, which heavily relies on the given trigger point and the proxy causal graph.

### G.3. BRCD with Nonparametric Estimators

As the theoretical guarantees are nonparametric, we provide an experiment to examine the performance of **BRCD** using a nonparametric estimator. We have used an extra tree regressor to group similar parent values to have the same group of samples with respect to their children to define a neighborhood of each observed value. We then use kernel density estimation with a Gaussian kernel to estimate the distribution based on the neighborhood for each observed value. We have tried it on Sockshop data and found that the performance has dropped, as shown by Table 8. Although our theory is grounded on nonparametric methods to obtain consistency, there can be some practical challenges in using nonparametric methods. For example, the nonparametric method we use here depends on how good the neighbors are, and that can be influenced by choosing different hyperparameters for the tree model. We only picked the default values in `scikit-learn`. Our theory only requires an $\varepsilon$-accurate likelihood approximation, not a specific parametric form. Thus, the guarantees are agnostic to the estimator and depend on approximation quality rather than model class. The linear Gaussian model is used

| | | CIRCA | RCG | BARO | RCD | IDI | ST | SO | CausalRCA | Cholesky | BRCD-C | BRCD-B10 | BRCD | SimpleRCA | ShapleyIQ | MircoDig |
|---|---|---|---|---|---|---|---|---|---|---|---|---|---|---|---|---|
| Avg@1 | CPU | 0.68 | 0.60 | **0.92** | 0.56 | 0.00 | 0.24 | 0.16 | 0.20 | 0.16 | **0.92** | 0.84 | 0.80 | **1.00** | 0.00 | 0.12 |
| | MEM | 0.72 | 0.36 | 0.92 | 0.28 | 0.00 | 0.12 | 0.20 | 0.20 | 0.08 | **0.96** | **0.96** | **0.96** | **1.00** | 0.00 | 0.04 |
| | DISK | 0.80 | 0.68 | 0.84 | 0.56 | 0.00 | 0.12 | 0.08 | 0.20 | 0.20 | **0.92** | 0.84 | 0.84 | **0.92** | 0.00 | 0.20 |
| | DELAY | 0.76 | 0.24 | **0.92** | 0.28 | 0.00 | 0.12 | 0.24 | 0.20 | 0.04 | 0.76 | 0.64 | 0.68 | **1.00** | 0.00 | 0.00 |
| | LOSS | 0.56 | 0.20 | **0.64** | 0.20 | 0.00 | 0.16 | 0.12 | 0.20 | 0.12 | 0.56 | 0.56 | 0.56 | **0.80** | 0.00 | 0.04 |
| | Average | 0.70 | 0.42 | **0.85** | 0.38 | 0.00 | 0.15 | 0.16 | 0.20 | 0.12 | 0.82 | 0.77 | 0.77 | **0.94** | 0.00 | 0.08 |
| Avg@3 | CPU | 0.89 | 0.75 | **0.97** | 0.65 | 0.00 | 0.24 | 0.31 | 0.33 | 0.24 | 0.96 | 0.92 | 0.93 | **1.00** | 0.00 | 0.32 |
| | MEM | 0.83 | 0.71 | **0.97** | 0.39 | 0.00 | 0.12 | 0.29 | 0.33 | 0.16 | **0.97** | 0.96 | **0.97** | **1.00** | 0.00 | 0.24 |
| | DISK | 0.87 | 0.84 | 0.93 | 0.63 | 0.00 | 0.12 | 0.25 | 0.33 | 0.35 | **0.97** | 0.92 | 0.91 | **0.97** | 0.00 | 0.35 |
| | DELAY | 0.91 | 0.57 | **0.97** | 0.36 | 0.00 | 0.12 | 0.33 | 0.33 | 0.11 | 0.81 | 0.77 | 0.75 | **1.00** | 0.00 | 0.20 |
| | LOSS | 0.72 | 0.49 | **0.83** | 0.28 | 0.00 | 0.16 | 0.24 | 0.33 | 0.23 | 0.72 | 0.68 | 0.67 | **0.89** | 0.00 | 0.23 |
| | Average | 0.84 | 0.67 | **0.93** | 0.46 | 0.00 | 0.15 | 0.28 | 0.33 | 0.22 | 0.89 | 0.85 | 0.85 | **0.97** | 0.00 | 0.27 |
| Avg@5 | CPU | 0.94 | 0.85 | **0.98** | 0.70 | 0.00 | 0.24 | 0.40 | 0.48 | 0.33 | **0.98** | 0.95 | 0.96 | **1.00** | 0.00 | 0.46 |
| | MEM | 0.89 | 0.82 | **0.98** | 0.48 | 0.00 | 0.12 | 0.43 | 0.48 | 0.31 | **0.98** | 0.96 | **0.98** | **1.00** | 0.00 | 0.46 |
| | DISK | 0.92 | 0.90 | 0.96 | 0.67 | 0.00 | 0.12 | 0.41 | 0.48 | 0.46 | **0.98** | 0.95 | 0.94 | **0.98** | 0.00 | 0.38 |
| | DELAY | 0.94 | 0.70 | **0.98** | 0.45 | 0.00 | 0.12 | 0.42 | 0.48 | 0.26 | 0.86 | 0.82 | 0.79 | **1.00** | 0.00 | 0.46 |
| | LOSS | 0.82 | 0.63 | **0.89** | 0.33 | 0.00 | 0.16 | 0.38 | 0.48 | 0.33 | 0.77 | 0.74 | 0.73 | **0.93** | 0.00 | 0.34 |
| | Average | 0.90 | 0.78 | **0.96** | 0.53 | 0.00 | 0.15 | 0.41 | 0.48 | 0.34 | 0.91 | 0.88 | 0.88 | **0.98** | 0.00 | 0.37 |

*Table 5.* Avg@k performance on RCAEval Sockshop RE1 dataset (Pham et al., 2025).

| | CIRCA | RCG | BARO | RCD | IDI | ST | SO | CausalRCA | Cholesky | BRCD-C | BRCD-B10 | BRCD | SimpleRCA | ShapleyIQ | MircoDig |
|---|---|---|---|---|---|---|---|---|---|---|---|---|---|---|---|
| OB | 0.62 | 0.68 | **0.84** | 0.58 | 0.48 | 0.33 | 0.25 | 0.37 | 0.28 | 0.81 | 0.74 | 0.73 | 0.78 | 0.72 | 0.42 |
| SP | 0.83 | 0.65 | 0.92 | 0.46 | 0.00 | 0.15 | 0.37 | 0.41 | 0.31 | 0.89 | 0.84 | 0.84 | **0.97** | 0.00 | 0.32 |

*Table 6.* Averaged Mean Reciprocal Rank across all failure types on RCAEval Online Boutique (OB) and Sockshop (SP) RE1 dataset (Pham et al., 2025).

| RCD | RCG | BARO | BRCD | BRCD-C | BRCD-M | BRCD-B100 | SO | ST | Cholesky | IDI | ShapleyIQ | MicroDig | SimpleRCA |
|---|---|---|---|---|---|---|---|---|---|---|---|---|---|
| 0.26 | 0.20 | 0.32 | 0.51 | 0.57 | 0.53 | 0.52 | 0.08 | 0.07 | 0.07 | 0.27 | 0.26 | 0.29 | 0.25 |

*Table 7.* **Averaged Mean Reciprocal Rank performance for the Petshop dataset** (Hardt et al., 2023). We see that the proposed algorithm **BRCD** and its variants (**BRCD-C**, **BRCD-M**, **BRCD-100**) perform the best. Each cell is an average MRR across all cases. There are a total of 26, 26, 8, and 8 datasets for high_traffic, low_traffic, temporal_traffic1, and temporal_traffic2, respectively. Each dataset contains only 5 anomalous samples.

| Failure Type | Avg@1 | Avg@3 | Avg@5 |
|---|---|---|---|
| CPU | 0.72 | 0.80 | 0.84 |
| MEM | 0.76 | 0.81 | 0.84 |
| DISK | 0.60 | 0.61 | 0.64 |
| DELAY | 0.60 | 0.68 | 0.73 |
| LOSS | 0.52 | 0.55 | 0.61 |
| **Average** | **0.64** | **0.69** | **0.73** |

*Table 8.* Average performance by failure type in the Sockshop experiment for **BRCD** with an nonparametric estimator.

for scalability and stability, and works well in practice because **BRCD** relies on relative likelihood comparisons, which are often preserved under mild misspecification.

