# OpenReview forum: "Root Cause Analysis of Failures in Microservices via Bayesian Root Cause Discovery"
_ICML.cc/2026/Conference — ICML 2026 spotlight_

### Official Review · Reviewer_68ap · 2026-03-06

**Soundness:** 3
**Presentation:** 3
**Significance:** 4
**Originality:** 2
**Overall Recommendation:** 4
**Confidence:** 3

**Summary:**

The paper proposes BRCD (Bayesian Root Cause Discovery), a method for identifying root causes of failures in microservice systems. The core idea is to use a partial causal structure (CPDAG) learned from pre-failure data, then perform Bayesian inference over root cause candidates using a polynomial-time DAG sampling algorithm rather than enumerating all DAGs.

The system failure is modeled as a soft intervention, represented by a binary proxy variable $F$ that points to the root cause in an augmented DAG. The algorithm avoids enumerating all possible DAGs by grouping them into Interventional Markov Equivalence Classes (I-MECs). All graphs within the same I-MEC entail the same data likelihood, allowing for efficient posterior updates.

The paper provides the first statistical consistency guarantees for nonparametric RCA, including identifiability of root causes and finite-sample posterior bounds.

**Compliance With Llm Reviewing Policy:**

Affirmed.

**Key Questions For Authors:**

1. The framework assumes causal sufficiency (no latent variables). In practice, microservice performance is often influenced by unobserved factors such as network "jitter" or shared hardware resource contention. How does BRCD's posterior behave when a root cause is an unmeasured latent variable?

2.  Microservice architectures frequently change due to deployments and autoscaling. How would BRCD handle scenarios where the causal structure shifts between the pre-failure CPDAG learning phase and the actual failure?

3.  The theoretical guarantees are nonparametric, but real-world experiments rely on a linear Gaussian likelihood estimator. How much of the empirical performance gain is attributable to the Bayesian framework itself versus the choice of likelihood estimator? Have the authors tested nonparametric likelihood estimators on the real-world datasets?

**Limitations:**

* Causal sufficiency is a strong assumption that is rarely justified in production microservice environments, where shared hardware, network infrastructure, and unmonitored third-party services routinely act as hidden common causes. The paper should discuss the expected degradation under latent confounding.

* The CPDAG learning phase is treated as essentially free, but in practice observational causal discovery on high-dimensional time series data is itself expensive and error-prone, especially under non-stationarity. The end-to-end pipeline cost is underreported.

* Real-world evaluation datasets are relatively small and curated. Petshop has only 68 injected faults, and OB/Sockshop experiments use datasets with at most a few thousand samples. Performance on larger-scale, messier production traces remains unknown.

**Strengths And Weaknesses:**

# Strengths
**Novelty:** The transition from constraint-based methods (which suffer from error propagation in finite samples) to a Bayesian approach using efficient DAG sampling is a significant advancement.

**Theoretical Rigor:** Providing identifiability and consistency proofs for nonparametric RCA is a strong contribution.

**Practicality:** The "anytime property" allows for iterative updates as more data arrives, which is crucial for real-time incident response.

# Weaknesses
**CPDAG Dependency:** The performance of BRCD is heavily dependent on the quality of the initial CPDAG. If the pre-failure causal discovery is inaccurate, the root cause ranking may suffer.

**Scalability:** While it scales to 1,000 nodes, microservice topologies are often highly dynamic. The paper assumes the causal structure is learned prior to failure; it's unclear how the model handles rapid service changes.

**No Latent Variables:** The assumption of no latent variables (causal sufficiency) is often violated in real-world IT environments where many hidden factors can influence service metrics.

**Linear Gaussian likelihood assumption:** Despite claiming to be nonparametric, the real-world experiments use a linear Gaussian likelihood estimator. This is a meaningful gap between theory and practice that deserves more discussion.

---

> ### Author Rebuttal · Authors · 2026-03-30
>
> We thank the reviewer for the thoughtful questions.
>
> 1. If the root cause is an unmeasured latent variable and there is no other unobserved confounder, then BRCD will be able to capture such a case by labeling the observed child as the root cause. In the case of having other unobserved confounders, this violates the assumption of the causal discovery algorithm we use for learning the CPDAG. We believe this is an important direction to extend BRCD with partial ancestral graphs (PAG) that relax causal sufficiency. However, updating the likelihood by sampling from PAGs efficiently will be a very challenging task. We examine how sensitive the performance of BRCD is to the presence of latent confounders. We follow a similar setup in our synthetic experiment. We generate 50 different DAGs per graph size $10$ with an additional $5$ more unobserved confounders added to the DAGs and $2n$ many edges. The root cause is randomly picked among the observed variables. Each node has $4$ states. We generate 10k observational samples with $100$ anomalous samples. Here, BRCD uses the learned CPDAG based on BeDu score based on the BOSS causal discovery algorithm from the observed data. RCG also uses the 0-order essential graph.  We see that BRCD still outperforms RCD and RCG in the presence of latent variables, even with a learned CPDAG from observed data.
>
>
> | Metric | BRCD              | RCD               | RCG               |
> |--------|-------------------|-------------------|-------------------|
> | Top-1  | 0.60 ± 0.07       | 0.50 ± 0.07       | 0.36 ± 0.06       |
> | Top-3  | 0.80 ± 0.06       | 0.64 ± 0.07       | 0.60 ± 0.06       |
> | Top-5  | 0.90 ± 0.04       | 0.74 ± 0.06       | 0.74 ± 0.06       |
>
> 2. BRCD assumes the causal structure remains the same between the pre-failure CPDAG learning phase and the actual failure. If the structure shifts between two phases, it will adversely affect BRCD, as the learned CPDAG may no longer be accurate. One will need to draw on the causal discovery algorithms from non-stationary or semi-stationary time-series [1, 2]. Also, the likelihood must account for temporal dependence, which complicates the plug-in estimation used in BRCD.
> - [1] S. Gao, R. Addanki, T. Yu, R. A. Rossi, M. Kocaoglu, “Causal Discovery in Semi-Stationary Time Series,” to appear in Proc. of NeurIPS’23, New Orleans, LA, USA, Dec. 2023.
> - [2] Mameche, Sarah, et al. "Spacetime: Causal discovery from non-stationary time series." Proceedings of the AAAI Conference on Artificial Intelligence. Vol. 39. No. 18. 2025.
>
> 3. Thank you for the insightful question.  We have used an extra tree regressor to group similar parent values to have the same group of samples with respect to their children to define a neighborhood of each observed value. We then use kernel density estimation with a Gaussian kernel to estimate the distribution based on the neighborhood for each observed value. We have tried it on Sockshop data and found that the performance has dropped, as shown below. Although our theory is grounded on nonparametric methods to obtain consistency, there can be some practical challenges in using nonparametric methods. For example, the nonparametric method we use here depends on how good the neighbors are, and that can be influenced by choosing different hyperparameters for the tree model.  We only picked the default values in scikit-learn. Our theory only requires an $\epsilon$-accurate likelihood approximation, not a specific parametric form. Thus, the guarantees are agnostic to the estimator and depend on approximation quality rather than model class. The linear Gaussian model is used for scalability and stability, and works well in practice because BRCD relies on relative likelihood comparisons, which are often preserved under mild misspecification.
>
> | Failure Type | Avg@1 | Avg@3 | Avg@5 |
> |----------|-------|-------|-------|
> | CPU      | 0.72  | 0.80  | 0.84  |
> | MEM      | 0.76  | 0.81  | 0.84  |
> | DISK     | 0.60  | 0.61  | 0.64  |
> | DELAY    | 0.60  | 0.68  | 0.73  |
> | LOSS     | 0.52  | 0.55  | 0.61  |

---

> > ### Author Rebuttal · Reviewer_68ap · 2026-04-06
> >
> > Thank you to the authors for their response. My concerns have been addressed.

---

### Official Review · Reviewer_h6ad · 2026-03-06

**Soundness:** 3
**Presentation:** 3
**Significance:** 3
**Originality:** 3
**Overall Recommendation:** 5
**Confidence:** 4

**Summary:**

The paper proposes a Bayesian approach to root cause analysis in microservices that leverages a CPDAG learned from pre-failure observational data and models failure as a soft intervention. It then identifies the root cause by enumerating interventional Markov equivalence classes (I-MECs) that uniquely identify the node on which the intervention has occurred. The approach has been evaluated on synthetic and real-world benchmarks.

**Compliance With Llm Reviewing Policy:**

Affirmed.

**Final Justification:**

The authors addressed all my questions, I keep my recommendation for acceptance.

**Key Questions For Authors:**

Overall, the paper is nicely written and provides a novel approach. I only have a few minor concerns/questions:

- It seems BRCD was given the ground-truth CPDAG in synthetic experiments while baselines were not. Would the gains in BRCD also persist with a learned CPDAG?
- Since observational samples do not contribute to discriminating between root causes sharing the same MEC, shouldn't the convergence rate be characterized in terms of $n_{\text{int}}$ instead?
- At first glance, BRCD seems to be extendable to temporal dependencies as well (e.g., important in metrics from microservices). Is there any particular assumption that would make an extension to time-series data difficult?

**Limitations:**

Yes

**Strengths And Weaknesses:**

Strengths:
- Interesting and novel approach
- Efficient perspective
- Theoretically sound
- Good related literature research

Weaknesses:
- Some of the theoretical contributions are incremental, but the overall approach remains novel
- Unfair baseline comparison in the experiments (minor)

---

> ### Author Rebuttal · Authors · 2026-03-30
>
> We sincerely thank the reviewer for the thoughtful questions.
>
> 1. For the synthetic experiments, we use the ground-truth CPDAG to validate the theoretical properties of BRCD without confounding effects from structure learning errors. Scaling the learned CPDAG from discrete data is challenging for the synthetic experiment, as we use the BOSS algorithm, which uses BIC for model selection; switching to discrete BDeu scores does not seem to scale equally fast. Alternatively, motivated by Reviewer 68ap, we have additionally conducted an experiment where we use a learned CPDAG in the presence of latent confounders to help understand the performance better without a ground-truth CPDAG.
>
> We follow a similar setup in our synthetic experiment. We generate $50$ different DAGs per graph size $10$, with an additional $5$ more unobserved confounders added to the DAGs and $2n$ many edges. The root cause is randomly picked among the observed variables. Each node has 4 states. We generate 10k observational samples with $100$ anomalous samples. Here, BRCD uses the learned CPDAG based on the BeDu score, based on the BOSS causal discovery algorithm from the observed data. RCG also uses the $0$-order essential graph. We see that BRCD still outperforms RCD and RCG in such a setting.
>
> | Metric | BRCD (Ours)             | RCD               | RCG               |
> |--------|-------------------|-------------------|-------------------|
> | Top-1  | 0.60 ± 0.07       | 0.50 ± 0.07       | 0.36 ± 0.06       |
> | Top-3  | 0.80 ± 0.06       | 0.64 ± 0.07       | 0.60 ± 0.06       |
> | Top-5  | 0.90 ± 0.04       | 0.74 ± 0.06       | 0.74 ± 0.06       |
>
> 2. Thank you for this insightful comment. We agree that, conditional on a fixed CPDAG, the information that discriminates among root cause candidates comes primarily from the interventional samples, since the observational distribution is shared across candidates within the same MEC. Our current theorem is stated in terms of the total sample size because it covers the more general setting where both (i) the partial causal structure may itself be estimated from observational data, and (ii) the likelihood is evaluated on the combined distribution with the proxy variable $F$. In that formulation, observational samples contribute indirectly through improving the CPDAG estimate and the plug-in likelihood approximation. We will clarify this distinction in the revision: observational samples mainly support structure learning and stable likelihood estimation, while anomalous samples drive the posterior concentration among competing root-cause candidates.
>
> 3. We agree that extending BRCD to temporal settings is a natural and important direction, especially for microservice metrics. The main challenge in extending to time-series data is that temporal dependencies violate the i.i.d. assumption and require modeling lagged causal relations. This introduces two main difficulties: 1. Structure learning complexity: Temporal graphs significantly enlarge the search space, making enumeration and DAG sampling more computationally expensive. 2. Likelihood estimation: The likelihood must account for temporal dependence, which complicates the plug-in estimation used in BRCD. We view extending BRCD to temporal causal models as a promising direction for future work.

---

> > ### Author Rebuttal · Reviewer_h6ad · 2026-04-03
> >
> > I want to thank the authors for their response. My questions were addressed and I keep my recommendation for acceptance.

---

### Official Review · Reviewer_yHtu · 2026-03-09

**Soundness:** 3
**Presentation:** 3
**Significance:** 3
**Originality:** 3
**Overall Recommendation:** 4
**Confidence:** 4

**Summary:**

This article's fundamental concept pertains to using Bayesian inference for root cause analysis (RCA) in microservice systems, leveraging partial causal structures learned from pre-failure observational data. The paper addresses the critical problem of identifying root causes of failures in complex microservice architectures, where service interdependencies make troubleshooting extremely challenging. Overall, the paper investigates the issue of sample-efficient root cause discovery in microservices when only a partial causal structure is available from pre-failure data, and limited post-failure intervention samples are obtainable.

**Compliance With Llm Reviewing Policy:**

Affirmed.

**Final Justification:**

The authors response solved my concerns. Therefore, I raised my score.

**Key Questions For Authors:**

1. There are only two comparison methods used in the paper, with many existing RCA methods ignored as shown in [1]. Can authors compare with more methods, maybe as shown in [1]?
2. Microservice systems typically have well-documented service architectures, call graphs, and network topologies. Or they can be derived from trace data. Why does the paper choose to learn a causal graph (CPDAG) entirely from observational data rather than leveraging the known physical network and service dependency information? Could you compare the performance of using learned CPDAGs versus using service-map-based causal graphs across all datasets, and discuss the trade-offs? Also discuss with existing methods using known graph as in [2]

[1] Rethinking the Evaluation of Microservice RCA with a Fault Propagation-Aware Benchmark. https://arxiv.org/pdf/2510.04711v2
[2] Shapleyiq: Influence quantification by shapley values for performance debugging of microservices. https://dl.acm.org/doi/pdf/10.1145/3623278.3624771

**Limitations:**

1. The comparison baselines are limited, and the evaluation is conducted on only a small number of datasets.

**Strengths And Weaknesses:**

S1: The paper provides rigorous theoretical guarantees, including identifiability, posterior consistency , and finite-sample bounds.
S2: BRCD demonstrates excellent performance with limited anomalous samples, which is crucial for real-world microservice incidents where failures are rare and costly.
S3: The framework is evaluated on three real-world microservice datasets (Petshop, Online Boutique, Sockshop) with promising results.

W1: The comparison focuses primarily on causal RCA methods (RCD, RCG, BARO), but lacks comparison with simpler statistical or learning-based approaches that might be more practical in some settings.
W2: Microservice systems typically have known service-level architectures and network topologies (e.g., service maps, known call dependencies), but the paper does not leverage this valuable structural information.

---

> ### Author Rebuttal · Authors · 2026-03-30
>
> We thank the reviewer for the thoughtful feedback.
>
> 1. We have added an evaluation metric known as MRR, as in [1], for all the real-world experiments. We also include three best-performing baselines besides the BARO algorithm in Table 5 from [1] : ShapleyIQ, SimpleRCA, and MicroDig, with modifications, as we note that SimpleRCA, ShapleyIQ, and MicroDig rely on trace data besides metric data. Our datasets do not contain any trace data. We report the overall averaged Mean Reciprocal Rank (MRR) for each algorithm along with SimpleRCA, MicroDig, and ShapleyIQ across all failure categories in Petshop, Sockshop and OB datasets below.  We observe that SimpleRCA has achieved a significant performance on the Sockshop and Online Boutique. As noted in [2], these two datasets oversimplify real-world failure scenarios, as SimpleRCA can use the 95th percentile of the normal data and take the difference between that and the maximum value from the anomalous data for each variable to find root causes. SimpleRCA underperforms in the Petshop data, which has a greater diversity of failure scenarios. MicroDig, and ShapleyIQ also underperform across the datasets.
>
> | Algorithm   | Sockshop | Online Boutique | Petshop |
> |-------------|----------|-----------------|---------|
> | CIRCA       | 0.8259   | 0.6219          | -       |
> | CausalRCA   | 0.4082   | 0.3689          | -       |
> | Cholesky    | 0.3101   | 0.2769          | 0.069   |
> | SO          | 0.3666   | 0.2519          | 0.078   |
> | ST          | 0.1520   | 0.3255          | 0.066   |
> | RCG         | 0.6476   | 0.6774          | 0.2022  |
> | RCD         | 0.4594   | 0.5816          | 0.2596  |
> | IDI         | 0.0000   | 0.4822          | 0.2666  |
> | BARO        | 0.9176   | **0.8387**          | 0.3204  |
> | BRCD-C  (Ours)    | 0.8852   | 0.8100          | **0.5711**       |
> | BRCD  (Ours)        | 0.8358   | 0.7339          | 0.5110  |
> | BRCD-10 (Ours)      | 0.8352   | 0.7393          | -       |
> | BRCD-100 (Ours)    | -        | -               | 0.5180  |
> | ShapleyIQ   | 0.0000   | 0.7186          | 0.2559    |
> | MicroDig    | 0.3259   | 0.4164          | 0.2945  |
> | SimpleRCA   | **0.9683**   | 0.7783          | 0.2525  |
>
> 2. The reason for learning CPDAGs is to give a flexible framework to incorporate observational causal discovery, as there can be uncertainty about the causal relations at the metric-level, even if there exists a call graph at the service level. The side information about the physical network and service dependency can be incorporated into BRCD. We have provided a version of BRCD that uses a service map as the proxy causal graph to alter the topological order among the observed metrics in a DAG and directly do a Bayesian update based on that DAG for the Sockshop and Online Boutique. It is called BRCD-C in Tables 2 & 3 in the Appendix. The performance has indeed improved. We have also tested BRCD-C on the Petshop. It also shows improvement over BRCD, even over BRCD-100, in top-1 accuracy. However, in some cases, a more data-driven approach may be advantageous, for example, BRCD-100 has a higher top5- accuracy in a low traffic scenario. We present the results below. We will include this observation in our manuscript. If one has absolute certainty about some causal relations among the observed metrics, they can be treated as background knowledge to guide the CPDAG learning process. For the BOSS algorithm, we can penalize the candidate DAGs that are inconsistent with the background knowledge and enforce the ancestral relations in the learned CPDAG. We call this version BRCD-M, as shown below. We see that BRCD-M is better than BRCD in the high_traffic, temporal_traffic_1, and temporal_traffic_2, but is worse than BRCD in the low_traffic in terms of top-1 accuracy.
>
> | Scenario            | BRCD-C (Top-1) | BRCD-C (Top-3) | BRCD-C (Top-5) | BRCD-M (Top-1) | BRCD-M (Top-3) | BRCD-M (Top-5) |
> |---------------------|----------------|----------------|----------------|----------------|----------------|----------------|
> | high_traffic        | 0.35           | 0.38           | 0.42           | 0.35           | 0.38           | 0.42           |
> | low_traffic         | 0.42           | 0.69           | 0.73           | 0.35           | 0.58           | 0.58           |
> | temporal_traffic1   | 0.62           | 0.75           | 0.75           | 0.62           | 0.75           | 0.75           |
> | temporal_traffic2   | 0.88           | 0.88           | 0.88           | 0.88           | 0.88           | 0.88           |
> | **Overall MRR**     | **0.5711**     | -              | -              | **0.5266**     | -              | -              |

---

> > ### Author Rebuttal · Reviewer_yHtu · 2026-04-02
> >
> > The authors response solved my concerns and I upgraded my scores.

---

### Official Review · Reviewer_sp9e · 2026-03-13

**Soundness:** 3
**Presentation:** 3
**Significance:** 3
**Originality:** 3
**Overall Recommendation:** 4
**Confidence:** 3

**Summary:**

The paper studies RCA of failures in microservice systems and proposes BRCD. The method learns a CPDAG from observational data before failure and constructs possible I-CPDAGs by introducing a failure indicator that points to the root cause. It estimates the likelihood of anomaly data by sampling DAGs from each I-CPDAG and ranks root cause candidates using posterior probabilities. The evaluation includes synthetic datasets and the Petshop microservice dataset, and the results show that BRCD achieves higher root cause identification accuracy than several baseline methods.

**Compliance With Llm Reviewing Policy:**

Affirmed.

**Final Justification:**

Thank you to the authors for the detailed responses. My concerns about the baseline comparison have been addressed and I will keep my positive score.

**Key Questions For Authors:**

1. Can you provide a clearer summary across all real-world datasets and discuss more explicitly in which settings BRCD does and does not outperform prior methods?
2. Why do you choose the clique-picking algorithm?
3. Can you provide additional discussion or experiments to show how the theoretical finite-sample bounds relate to the empirical performance, for example, by analyzing convergence behavior or the practical tightness of the bound through evaluation?
4. How sensitive is BRCD to the choice of prior? In particular, the paper seems to use a uniform prior over the I-CPDAG, but it is not clear whether alternative priors based on service topology or expert knowledge would change the results.

**Limitations:**

yes

**Strengths And Weaknesses:**

# Strengths and Weaknesses
- S1. This paper proposes a novel method that applies a Bayesian approach to address the lack of historical failure interventions in RCA.

- S2. The paper has a solid theoretical foundation, and the proposed method is well supported by theory.

- S3. The proposed BRCD algorithm achieves high efficiency for large-scale graphs.

- W1. The evaluation on real-world datasets is limited.
The empirical evaluation appears to rely heavily on synthetic experiments, while the main real-world results presented in the paper are limited to the Petshop dataset. Although the paper mentions additional experiments on Online Boutique and Sockshop, these results are deferred to the appendix, and the appendix results on other real-world benchmarks do not seem to show that BRCD consistently achieves state-of-the-art performance (compared with BARO). This makes the overall empirical advantage less clear than suggested by the main paper.

- W2. The presentation of the evaluation is somewhat confusing.
The organization of the experimental results could be improved for readability. In the current presentation, it is not always immediately clear which figures correspond to synthetic datasets and which results correspond to the Petshop dataset. For example, Figure 2 and Figure 3 report synthetic results, while Table 1 reports Petshop results, but this separation is not mentioned in the figure/table titles. A clearer organization of the evaluation part would make it easier to follow.

- W3. The use of clique-sampling in Algorithm 1 is insufficiently explained in the main text.
Algorithm 1 states that a DAG is sampled via the clique-picking algorithm, but the main text does not provide enough explanation of why clique-picking is the appropriate choice in this setting.

- W4. The theoretical finite-sample error bounds are not empirically validated.
The paper provides theoretical results on posterior consistency and finite-sample bounds, which are presented as an important part of the contribution. However, the experimental section does not seem to directly evaluate or discuss these bounds. For example, the evaluation does not compare empirical behavior against the predicted finite-sample trend, nor does it examine whether the bound is informative or tight in practice in real-world data.

---

> ### Author Rebuttal · Authors · 2026-03-30
>
> Thank you for the questions. We answer them below.
>
> 1. Across all real-world experiments (Petshop, Online Boutique, and Sockshop), BRCD consistently achieves the best or near-best top-l accuracy, with the most notable gains in low-sample regimes.
> - Petshop (Table 1): BRCD and its bootstrapped variants outperform all baselines on average across traffic scenarios, despite having only 5 anomalous samples per dataset.
> - Online Boutique and Sockshop (Table 2 and 3 in the Appendix): following BARO, BRCD also achieves near-best performance, particularly when leveraging learned CPDAGs, confirming its scalability and effectiveness in larger systems.
> - Where BRCD performs best: When anomalous samples are limited, and the normal samples are sufficient to learn a good CPDAG. In moderate-to-large graphs, the DAG sampling avoids the error propagation seen in constraint-based methods.
> - Where gains are smaller: When the learned CPDAG is highly inaccurate due to a limited *normal* sample size, this is evident in the DISK, DELAY, and LOSS failure types as compared to CPU and MEM in the Online Boutique and Sockshop datasets.
>
> 2. We choose the clique-picking algorithm (Wienöbst et al., 2023) because it enables efficient and unbiased sampling of DAGs from a CPDAG in polynomial time, which is critical for making Bayesian inference tractable in our setting. In BRCD, we need to repeatedly sample DAGs from each I-MEC to approximate $p(\mathcal{D}|\mathbf{R})$. Without such a sampler, enumerating all DAGs would be computationally infeasible due to the super-exponential size of the equivalence class. This description is provided in the paragraph under Algorithm 2 in Section 4.
>
> 3. Thank you for your suggestion. The finite-sample bound in Theorem 4.4 shows that the posterior on the true root cause converges to 1 at an exponential rate in the number of samples, governed by the effective gap $\delta_{eff} = \delta_{\min} - 2B\epsilon$. Empirically, this behavior is reflected in our results. In particular, Figure 3 shows that BRCD’s top-l accuracy improves rapidly with increasing anomalous sample size, reaching strong performance with as few as 100 samples. This aligns with the theory’s prediction of fast concentration when the effective gap is positive. Similarly, the robustness under very limited anomalous samples (e.g. 5 samples in Petshop) is consistent with the bounds’ dependence on the KL separation rather than large sample sizes. We note that the bound is primarily qualitative rather than tight in practice, as it depends on unknown quantities such as $\delta_{\min}$ and the approximation error $\epsilon$. Directly estimating these terms is challenging, which limits precise numerical validation. However, the observed convergence trends are consistent with the theoretical guarantees.
>
> 4. As shown by Table 2 and Table 3 in the Appendix, we present two variants of BRCD (BRCD-C: uses call graph to enforce ancestral relations at the service-level among metrics, BRCD-10: takes 10 bootstrapping samples to learn CPDAGs and incorporates this as CPDAG prior for BRCD). BRCD-C uses the service map as the proxy causal graph as an alternative to the uniform prior. We can see the improvement over BRCD, which uses a uniform prior for all DAGs in the given CPDAG. In response to Reviewer yHtu, we also provide an additional version that takes the service map side information as background knowledge in BOSS algorithm to guide the CPDAG learning. Overall, directly incorporating a DAG (assuming there is no relation among metrics within the same service) for BRCD, namely BRCD-C, has achieved the best results.

---

> > ### Author Rebuttal · Reviewer_sp9e · 2026-04-04
> >
> > Thank you for the rebuttal. It addresses my concerns about the algorithm choice, the discussion of the finite-sample bound, and the prior. However, my concern about the real-world comparison between BRCD and the baselines is not fully resolved, and I believe the corresponding claims in the paper should be revised to better match the empirical results. I will keep my original score.

---

> > > ### Author Response · Authors · 2026-04-05
> > >
> > > We thank the reviewer for the feedback. We believe the mismatch the reviewer refers to concerns the statement: “Across OB, Sockshop, and Petshop (Tables 1–3), BRCD achieves the best average performance, even when the ground-truth CPDAG is unavailable.”
> > >
> > > Our intended meaning was that BRCD performs best overall: it achieves near-best performance on the OB and Sockshop datasets and the best performance on the Petshop dataset. We agree that the current phrasing may be misleading and will revise it to more accurately reflect the empirical results.
> > >
> > > Please let us know if this clarification does not address your concern.

---

### Decision · Program_Chairs · 2026-04-30

**Decision:**

Accept (spotlight)

**Comment:**

The paper presents a novel and technically sound Bayesian framework for root cause analysis in microservice systems, combining an interesting methodological idea with solid theoretical support and sufficient empirical evidence. The reviewers are broadly positive about the originality, soundness, and potential impact of the approach. The rebuttal further strengthened the paper and its contribution by clarifying the role of clique-based DAG sampling, expanding empirical comparisons, and addressing questions about priors, latent confounding, and the use of service-map information. I believe this can be a great addition to ICML’s program, and I recommend it for publication.

As minor suggestions for the final version, the authors should slightly moderate and sharpen the wording around real-world empirical performance so that the claims match the evidence precisely.